# ZERO-SHOT TRANSFER LEARNING FOR GRAY-BOX HYPER-PARAMETER OPTIMIZATION

## ABSTRACT

Zero-shot hyper-parameter optimization refers to the process of selecting hyper-parameter configurations that are expected to perform well for a given dataset upfront, without access to any observations of the losses of the target response. Existing zero-shot approaches are posed as initialization strategies for Bayesian Optimization and they often rely on engineered meta-features to measure dataset similarity, operating under the assumption that the responses of similar datasets behaves similarly with respect to the same hyper-parameters. Solutions for zero-shot HPO are embarrassingly parallelizable and thus can reduce vastly the required wallclock time of learning a single model. We propose a very simple HPO model called **Gr**ay-box Zero(**O**)-**S**hot **I**nitialization (GROSI) as a conditional parametric surrogate that learns a universal response model by exploiting the relationship between the hyper-parameters and the dataset meta-features directly. In contrast to existing HPO solutions, we achieve transfer of knowledge without engineered meta-features, but rather through a shared model that is trained simultaneously across all datasets. We design and optimize a novel loss function that allows us to regress from the dataset/hyper-parameter pair unto the response. Experiments on 120 datasets demonstrate the strong performance of GROSI, compared to conventional initialization strategies. We also show that by fine-tuning GROSI to the target dataset, we can outperform state-of-the-art sequential HPO algorithms.

## 1    INTRODUCTION

Within the research community, the concentration of efforts towards solving the problem of hyper-parameter optimization (HPO) has been mainly through sequential model-based optimization (SMBO), i.e. iteratively fitting a probabilistic response model, typically a Gaussian process (Rasmussen (2003)), to a history of observations of losses of the target response, and suggesting the next hyper-parameters via a policy, acquisition function, that balances exploration and exploitation by leveraging the uncertainty in the posterior distribution (Jones et al. (1998); Wistuba et al. (2018); Snoek et al. (2012)). However, even when solutions are defined in conjunction with transfer learning techniques (Bardenet et al. (2013); Wistuba et al. (2016); Feurer et al. (2015)), the performance of SMBO is heavily affected by the choice of the initial hyper-parameters. Furthermore, SMBO is sequential by design and additional acceleration by parallelization is not possible.

In this paper, we present the problem of zero-shot hyper-parameter optimization as a meta-learning objective that exploits dataset information as part of the surrogate model. Instead of treating HPO as a black-box function, operating blindly on the response of the hyper-parameters alone, we treat it as a gray-box function (Whitley et al. (2016)), by capturing the relationship among the dataset meta-features and hyper-parameters to approximate the response model.

In this paper, we propose a novel formulation of HPO as a conditional gray-box function optimization problem, Section 4, that allows us to regress from the dataset/hyper-parameter pair directly onto the response. Driven by the assumption that similar datasets should have similar response approximations, we introduce an additional data-driven similarity regularization objective to penalize the difference between the predicted response of similar datasets. In Section 5, we perform an extensive battery of experiments that highlight the capacity of our universal model to serve as a solution for: (1) zero-shot HPO as a stand-alone task, (2) zero-shot as an initialization strategy for Bayesian Optimization (BO), (3) transferable sequential model-based optimization. A summary of our contributions is:

- a formulation of the zero-shot hyper-parameter optimization problem in which our response model predicts upfront the full set of hyper-parameter configurations to try, without access to observations of losses of the target response;

- a novel multi-task optimization objective that models the inherent similarity between datasets and their respective responses;

- three new meta-datasets with different search spaces and cardinalities to facilitate the experiments and serve as a benchmark for future work;

- an empirical demonstration of the performance of our approach through a battery of experiments that address the aforementioned research aspects, and a comparison against state-of-the-art HPO solutions for transfer-learning.

## 2 RELATED WORK

The straightforward zero-shot approaches for HPO consist of random search (Bergstra & Bengio (2012)), or simply selecting hyper-parameters that perform well on general tasks (Brazdil et al. (2003)). Some recent work has also shown that simply selecting random hyper-parameters from a restricted search space significantly outperforms existing solutions, and improves the performance of conventional SMBO approaches (Perrone et al. (2019)). The restricted search space is created by eliminating regions that are further away from the best hyper-parameters of the training tasks.

Another prominent direction for zero-shot HPO depends heavily on engineered meta-features, i.e. dataset characteristics (Vanschoren (2018)), to measure the similarity of datasets. Following the assumption that the responses of similar datasets behave similarly to the hyper-parameters, it has been shown that even the simplest of meta-features (Bardenet et al. (2013)) improve the performance of single task BO algorithms (Feurer et al. (2014; 2015)). The target response is initialized with the top-performing hyper-parameters of the dataset nearest neighbor in the meta-feature space. The shortcomings of using engineered meta-features are that they are hard to define (Leite & Brazdil (2005)), and are often selected through trial-and-error or expert domain knowledge. As a remedy, replacing engineered meta-features with learned meta-features (Jomaa et al. (2019)) compensates for such limitations, by producing expressive meta-features agnostic to any meta-task, such as HPO.

Zero-shot HPO is also posed as an optimization problem that aims to minimize the meta-loss over a collection of datasets (Wistuba et al. (2015a)) by replacing the discrete minimum function with a differentiable softmin function as an approximation. The initial configurations boost the single task BO without any meta-features. In (Wistuba et al. (2015b)), hyper-parameter combinations are assigned a static ranking based on the cumulative average normalized error, and dataset similarity is estimated based on the relative ranking of these combinations. Winkelmolen et al. (2020) introduce a Bayesian Optimization solution for zero-shot HPO by iteratively fitting a surrogate model over the observed responses of different tasks, and selecting the next hyper-parameters and datasets that minimize the aggregated observed loss.

Aside from zero-shot HPO, transfer learning is employed by learning better response models (Wistuba et al. (2016)) based on the similarity of the response. Feurer et al. (2018) propose an ensemble model for BO by building the target response model as a weighted sum of the predictions of base models as well as the target model. In addition to the transferable response models, Volpp et al. (2019) design a transferable acquisition function as a policy for hyper-parameter optimization defined in a reinforcement learning framework. As a replacement to the standard Gaussian process, Perrone et al. (2018) train a multi-task adaptive Bayesian linear regression model with a shared feature extractor that provides context information for each independent task.

In contrast to the literature, we formulate the problem of zero-shot HPO as a gray-box function optimization problem, by designing a universal response model defined over the combined domain of datasets and hyper-parameters. We rely on the embeddings to estimate the similarities across datasets and design a novel multi-task optimization objective to regress directly on the response. This allows us to delineate from the complexity paired with Bayesian uncertainty, as well as the trouble of engineering similarity measures.

## 3 HYPER-PARAMETER OPTIMIZATION

Consider a dataset $D = \left\{ \left( x^{(\text{Train})}, y^{(\text{Train})} \right), \left( x^{(\text{Val})}, y^{(\text{Val})} \right), \left( x^{(\text{Test})}, y^{(\text{Test})} \right) \right\}$ for a supervised learning task, with training, validation and test splits of predictors $x \in \mathcal{X}$ and targets $y \in \mathcal{Y}$. We aim at training a parametric approximation of the target using $\hat{y} := f(\theta, \lambda) : \mathcal{X} \to \mathcal{Y}$, where $\theta \in \Theta$ denotes the parameters and $\lambda \in \Lambda$ its hyper-parameters, by minimizing a loss function $\mathcal{L} : \mathcal{Y} \times \mathcal{Y} \to \mathbb{R}$ as:

$$\lambda^* = \underset{\lambda \in \Lambda}{\arg\min} \ \mathcal{L}\left( y^{(\text{Val})}, f\left( x^{(\text{Val})}; \theta^*, \lambda \right) \right) \quad \text{s.t.} \quad \theta^* = \underset{\theta \in \Theta}{\arg\min} \ \mathcal{L}\left( y^{(\text{Train})}, f\left( x^{(\text{Train})}; \theta, \lambda \right) \right) \quad (1)$$

We hereafter denote the validation error as the response $\ell(\lambda) := \mathcal{L}\left( y^{(\text{Val})}, f\left( x^{(\text{Val})}; \theta^*, \lambda \right) \right)$. Unfortunately, a direct optimization of the response $\ell(\lambda)$ in terms of $\lambda$ is not trivial, because $\theta^*$ is the result of the minimization problem and its gradients with respect to $\lambda$ are not easy to compute. Instead, in order to learn the optimal hyper-parameters $\lambda$ we train a probabilistic surrogate $\hat{\ell}(\lambda; \beta) : \Lambda \times \mathcal{B} \to R$ parameterized by $\beta \in \mathcal{B}$, with $\mathcal{B}$ as the space of response model parameters, that minimizes the log-likelihood of approximating the response $\ell(\lambda)$ over a set of $K$ evaluations $S := \{(\lambda_1, \ell(\lambda_1)), \dots, (\lambda_K, \ell(\lambda_K))\}$. We denote $P$ as the probability of estimating the response given a surrogate model. Given the surrogate, the next hyper-parameter to be evaluated $\lambda^{(\text{next})}$ is computed by maximizing an acquisition function $\mathcal{A}$ (e.g. EI (Močkus (1975))) as:

$$\lambda^{(\text{next})} := \underset{\lambda \in \Lambda}{\arg\max} \ \mathcal{A}(\hat{\ell}(\lambda; \beta^*)) \quad \text{s.t.} \quad \beta^* := \underset{\beta \in \mathcal{B}}{\arg\min} \ \sum_{k=1}^{K} \ln P\left( \ell(\lambda_k), \ \hat{\ell}(\lambda_k; \beta) \right) \quad (2)$$

## 4 META-LEARNING OF CONDITIONAL GRAY-BOX SURROGATES

Let us define a collection of $T$ datasets as $\left\{ D^{(1)}, \dots, D^{(T)} \right\}$ and let $\ell^{(t)}(\lambda)$ measure the response of the hyper-parameter $\lambda$ on the $t$-th dataset $D^{(t)}$. Furthermore, assume we have previously evaluated $K^{(t)}$ many hyper-parameters $\lambda_k^{(t)}, k \in \{1, \dots, K^{(t)}\}$ on that particular dataset. We condition the surrogate $\hat{\ell}$ to capture the characteristics of the $t$-th dataset, by taking as input the meta-features representation of the dataset as $\phi^{(t)}$. Therefore, a dataset-aware surrogate can be trained using meta-learning over a cumulative objective function $\mathcal{O}(\beta)$ as:

$$\mathcal{O}(\beta) \ := \ \sum_{t=1}^{T} \sum_{k=1}^{K^{(t)}} \left( \ell^{(t)}\left( \lambda_k^{(t)} \right) - \hat{\ell}\left( \lambda_k^{(t)}, \phi^{(t)}; \ \beta \right) \right)^2 \tag{3}$$

### 4.1 THE META-FEATURE EXTRACTOR

Introducing engineered meta-features has had a significant impact on hyper-parameter optimization. However, learning meta-features across datasets of varying schema in a task-agnostic setting provides more representative characteristics than to rely on hard-to-tune empirical estimates. The meta-feature extractor is a set-based function (Zaheer et al. (2017)) that presents itself as an extended derivation of the Kolmogorov-Arnold representation theorem (Krková (1992)), which states that a multi-variate function $\phi$ can be defined as an aggregation of univariate functions over single variables, Appendix B.

Each supervised (tabular) dataset $D^{(t)} := \left( x^{(t)}, y^{(t)} \right)$ consists of instances $x^{(t)} \in \mathcal{X} \in \mathbb{R}^{N \times M}$ and targets $y^{(t)} \in \mathcal{Y} \in \mathbb{R}^{N \times C}$ such that $N$, $M$ and $C$ represent the number of instances, predictors and targets respectively. The dataset can be further represented as a set of smaller components, *set of sets*, $D^{(t)} = \left\{ \left( x_{i,m}^{(t)}, y_{i,c}^{(t)} \right) \mid m \in \{1, \dots, M\}, i \in \{1, \dots, N\}, c \in \{1, \dots, C\} \right\}$. A tabular dataset composed of columns (predictors, targets) and rows (instances) is reduced to *single* predictor-target pairs instead of an instance-target pairs. Based on this representation, a meta-feature extractor parameterized as a neural network (Jomaa et al. (2019)), is formulated in Equation 4. For simplicity

of notation, we drop the superscript $(t)$ unless needed.

$$\phi(D) = h\left(\frac{1}{MC}\sum_{m=1}^{M}\sum_{c=1}^{C}g\left(\frac{1}{N}\sum_{i=1}^{N}f(x_{i,m}, y_{i,c})\right)\right) \tag{4}$$

with $f : \mathbb{R}^2 \to \mathbb{R}^{K_f}$, $g : \mathbb{R}^{K_f} \to \mathbb{R}^{K_g}$ and $h : \mathbb{R}^{K_g} \to \mathbb{R}^K$ represented by neural networks with $K_f$, $K_g$, and $K$ output units, respectively. This set-based formulation captures the correlation between each variable (predictor) and its assigned target and is permutation-invariant, i.e. the output is unaffected by the ordering of the pairs in the set. Other set-based functions such as (Edwards & Storkey (2016); Lee et al. (2019)) can also be used for meta-feature extraction, however, we focus on this deep-set formulation (Jomaa et al. (2019)) because it is proven to work properly for hyper-parameter optimization.

## 4.2 THE AUXILIARY DATASET IDENTIFICATION TASK

The dataset identification task introduced previously as dataset similarity learning (Jomaa et al. (2019)), ensures that the meta-features of *similar* datasets are colocated in the meta-feature space, providing more expressive and distinct meta-features for every dataset.

Let $p_D$ a joint distribution over *dataset* pairs such that $(D^{(t)}, D^{(q)}, s) \in T \times T \times \{0, 1\}$ with $s$ being a binary dataset similarity indicator. We define a classification model $\hat{s} : T \times T \to \mathbb{R}^+$ that provides an unnormalized probability estimate for $s$ being 1, as follows:

$$\hat{s}(D^{(t)}, D^{(q)}) = e^{-\gamma Z(\phi^{(t)}, \phi^{(q)})} \tag{5}$$

where $Z : \mathbb{R}^k \times \mathbb{R}^k \to \mathbb{R}^+$ represents any distance metric, and $\gamma$ a tuneable hyper-parameter. For simplicity, we use the Euclidean distance to measure the similarity between the extracted meta-features, i.e. $Z(\phi^{(t)}, \phi^{(q)}) = \|\phi^{(t)} - \phi^{(q)}\|$, and set $\gamma = 1$. The classification model is trained by optimizing the negative log likelihood:

$$\mathcal{P}(\beta) := -\sum_{(t,q)\sim p_{D+}}\log\left(\hat{s}(D^{(t)}, D^{(q)})\right) - \sum_{(t,q)\sim p_{D-}}\log\left(1 - \hat{s}(D^{(t)}, D^{(q)})\right) \tag{6}$$

with $p_{D+}$ as the distribution of similar datasets, $p_{D+} = \{(D^{(t)}, D^{(q)}, s) \sim p_D \mid s = 1\}$, and $p_{D-}$ as the distribution of dissimilar datasets, $p_{D-} = \{(D^{(t)}, D^{(q)}, s) \sim p_D \mid s = 0\}$. Similar datasets are defined as multi-fidelity subsets (batches) of each dataset.

## 4.3 DATA-DRIVEN SIMILARITY REGULARIZATION

Our surrogate differs from prior practices, because we do not consider the response to be entirely black-box. Instead, since we know the features and the target values of a dataset even before evaluating any hyper-parameter, we model a **gray-box** surrogate by exploiting the dataset characteristics $\phi$ when approximating the response $\ell$. As a result, if the surrogate faces a new dataset that is similar to one of the $T$ datasets from the collection it was optimized (i.e. similar meta-features $\phi$ extracted directly from the dataset), it will estimate a similar response. Yet, if we know apriori that two datasets are similar by means of the distance of their meta-features, we can explicitly regularize the surrogate to produce similar response estimations for such similar datasets, as:

$$\mathcal{R}(\beta) := \sum_{t=1}^{T-1}\sum_{q=t+1}^{T}\sum_{k=1}^{K^{(t)}}\|\phi^{(t)} - \phi^{(q)}\|\left(\hat{\ell}\left(\lambda_k^{(t)}, \phi^{(t)};\ \beta\right) - \hat{\ell}\left(\lambda_k^{(t)}, \phi^{(q)};\ \beta\right)\right)^2 \tag{7}$$

Overall we train the surrogate model to estimate the collection of response evaluations and explicitly capture the dataset similarity by solving the following problem, Equation 8, *end-to-end*, where $\alpha \in \mathbb{R}$ controls the amount of similarity regularization, and $\delta \in \mathbb{R}$ controls the impact of the dataset identification task:

$$\beta^* := \underset{\beta \in \mathcal{B}}{\arg\min}\ \ \mathcal{O}(\beta) + \alpha\,\mathcal{R}(\beta) + \delta\,\mathcal{P}(\beta) \tag{8}$$

NETWORK ARCHITECTURE

Our model architecture is divided into two modules, $\hat{l} := \phi \circ \psi$, the meta-feature extractor $\phi$, and the regression head $\psi$. The meta-feature extractor $\phi : \mathbb{R}^2 \to \mathbb{R}^{K_h}$ is composed of three functions, Equation 4, namely $f$, $g$ and $h$. The regression head is also composed of two functions, i.e. $\psi : \psi_1 \circ \psi_2$. We define by $\psi_1 : \mathbb{R}^{K_h} \times \Lambda \to \mathbb{R}^{K_{\psi_1}}$ as the function that takes as input the meta-feature/hyper-parameter pair, and by $\psi_2 : \mathbb{R}^{K_{\psi_1}} \to \mathbb{R}$ the function that approximates the response. Finally, let **Dense(n)** define one fully connected layer with $n$ neurons, and **ResidualBlock(n,m)** be $m \times$ Dense(n) with residual connections (Zagoruyko & Komodakis (2016)). We select the architecture presented in Table 1 based on the best observed average performance on the held-out validation sets across all meta-datasets, Appendix E.1.

Table 1: The network architecture optimized for every meta-dataset.

| Functions | Architecture |
|---|---|
| $f$ | Dense(16);6×ResidualBlock(3,16);Dense(16) |
| $g$ | Layout: ▷ with 3 layers and 16 Neurons |
| $h$ | Dense(16);3×ResidualBlock(3,16);Dense(16) |
| $\psi_1$ | Layout: ▷ with 4 layers and 4 Neurons |
| $\psi_2$ | Layout: ▷ with 4 layers and 16 Neurons |

## 5 EXPERIMENTS

Our experiments are designed to answer three research questions[1]:

- **Q1**: Can we learn a universal response model that provides useful hyper-parameter initializations from unseen datasets without access to previous observations of hyper-parameters for the dataset itself?

- **Q2**: Do the proposed suggestions serve as a good initialization strategy for existing SMBO algorithms?

- **Q3**: Aside from zero-shot HPO, does the performance of our method improve by refitting the response model to the observations of the hyper-parameters for the target dataset and how well does our approach compare to state-of-the-art methods in transfer learning for HPO?[2]

### 5.1 TRAINING PROTOCOL

In Algorithm 1 we describe the pseudo-code for optimizing our response model via standard meta-learning optimization routines. We use stochastic gradient descent to optimize the internal model, and Adam optimizer (Kingma & Ba (2015)) to optimize the outer loop. We set the number of inner iterations to $v = 5$, and use a learning rate of $0.001$ for both optimizers. We use a batch size of $8$ tasks sampled randomly with each iteration. The code is implemented in Tensoflow (Abadi et al. (2016)). The performance of the various optimizers is assessed by measuring the regret, which represents the distance between an observed response and the optimal response on a response surface. For hyper-parameter optimization, the meta-datasets are provided beforehand, consequently, the optimal response is known. Since we normalize the response surfaces between $(0, 1)$, we observe the *normalized* regret. The reported results represent the average over 5-fold cross-validation split for each meta-dataset, with 80 meta-train, 16 meta-valid, and 24 meta-test sets, and one unit of standard deviation.

### 5.2 META-DATASET

We create three meta-datasets by using 120 datasets chosen from the UCI repository (Asuncion & Newman (2007)). We then create the meta-instances by training a feedforward neural network

---

[1]For a better understanding of the different problem settings, see Appendix A

[2]The associated code and meta-dataset described will be available upon acceptance.

and report the validation accuracy. Each dataset is provided with a predefined split 60% train, 15% validation, and 25% test instances. We train each configuration for 50 epochs with a learning rate of 0.001. The hyper-parameter search space is described in Table 2.

Table 2: Hyper-parameter search space for the meta-datasets. The name of each the meta-datasets is inspired by the most prominent hyper-parameter, highlighted in red.

| Hyper-parameter | Layout Md | Regularization Md | Optimization Md |
|---|---|---|---|
| Activation | ReLU, SeLU | ReLU, SeLU, LeakyReLU | ReLU, SeLU, LeakyReLU |
| Neurons | $4, 8, 16, 32$ | $4, 8, 16, 32$ | $4, 8, 16$ |
| Layers | $1, 3, 5, 7$ | $1, 3, 5, 7$ | $3, 5, 7$ |
| Layout | $\square, \triangleleft, \triangleright, \diamond, \triangle$ | $\square$ | $\triangleleft, \triangleright, \diamond, \triangle$ |
| Dropout | $0, 0.5$ | $0, 0.2, 0.5$ | $0$ |
| Normalization | False | False, True | False |
| Optimizer | ADAM | ADAM | ADAM, RMSProp, GD |

The **layout** hyper-parameter (Jomaa et al. (2019)) corresponds to the overall shape of the neural network, and provides information regarding the number of neurons in each layer. For example, all the layers in the neural network with a $\square$ layout share the same number of **neurons**. We introduce an additional layout, $\triangle$, where the number of neurons in each layer is successively halved until it reaches the corresponding number of **neurons** in the central layer, then doubles successively. We also use dropout (Srivastava et al. (2014)) and batch normalization (Ioffe & Szegedy (2015)) as regularization strategies, and stochastic gradient descent (GD), ADAM (Kingma & Ba (2015)) and RMSProp (Tieleman & Hinton (2012)) as optimizers. SeLU (Klambauer et al. (2017)) represents the self-normalizing activation unit. The search space consists of all possible combinations of the hyper-parameters. After removing redundant configurations, the resulting meta-datasets have 256, 288 and 324 unique configurations respectively. For the purposes of our algorithm, we need access to the datasets used to generate the meta-features[3]. Further details are available in Appendix C.

## 5.3 BASELINES

We introduce two sets of baselines to evaluate against the different aspects of our approach:

ZERO-SHOT HYPER-PARAMETER OPTIMIZATION

- **Random** search (Bergstra & Bengio (2012)) is the simplest approach where the hyper-parameters are selected randomly.

- **Average Rank** represents the top hyper-parameters that had on average the highest-ranking across the meta-train datasets.

- **NN-⟨METAFEATURE⟩** (Feurer et al. (2015)) refers to the process of selecting the top-performing hyper-parameters of the nearest neighboring dataset based on their meta-features. We use two sets of well-established engineered meta-features, which we refer to as **MF1** (Feurer et al. (2015)) and **MF2** (Wistuba et al. (2016)), as well as learned meta-features (Jomaa et al. (2019)), which we denote by **D2V**. The similarity is measured by the Euclidean distance.

- **Ellipsoid** (Perrone et al. (2019)) is also a random search approach, however the hyper-parameters are sampled from a hyper-ellipsoid search space that is restricted to encompass as many optimal hyper-parameters from the training dataset as possible.

SEQUENTIAL-MODEL BASED OPTIMIZATION FOR TRANSFER LEARNING

- **GP** (Rasmussen (2003)) is standard Gaussian process response model with a Matern 3/2 and automatic relevance determination. This approach is trained independently on each dataset.

---

[3]Unfortunately, we could not evaluate our approach on some of the published meta-datasets (Schilling et al. (2016)) due to the unavailability of the associated datasets (original predictors and target values) used for generation of the meta-instances

- **SMFO** (Wistuba et al. (2015b)) is a sequential model-free approach that provides a collection of hyper-parameters by minimizing the ranking loss across all the tasks in the meta-train datasets.

- **TST-R** (Wistuba et al. (2016)) is a two-stage approach where the parameters of the target response model are adjusted via a kernel-weighted average based on the similarity of the hyper-parameter response between the target dataset and the training datasets. We also evaluate the variant of this approach that relies on meta-features, by replacing the engineered meta-features with learned meta-features, **TST-D2V**.

- **RGPE** (Feurer et al. (2018)) is an ensemble model that estimates the target response model as a weighted combination of the training datasets' response models and the target itself. The weights are assigned based on a ranking loss of the respective model.

- **ABLR** (Perrone et al. (2018)) is a multi-task ensemble of adaptive Bayesian linear regression models with all the tasks sharing a common feature extractor.

- **TAF-R** (Wistuba et al. (2018)) learns a transferable acquisition function, unlike the afore-mentioned algorithms that focus on a transferable response model, that selects the next hyper-parameter based on a weighted combination of the expected improvement of the target task, and predicted improvement on the source tasks.

- **MetaBO** (Volpp et al. (2019)) is another transferable acquisition function, optimized as a policy in a reinforcement learning framework. This approach, however, demands a pre-computed target response model as part of the state representation.

In our approach, we learn a universal response model based on the underlying assumption that the response is not only dependent on the hyper-parameters, as is assumed in black-box optimization techniques, but also on the dataset itself, presenting the problem as a gray-box function optimization.

### 5.4 RESULTS AND DISCUSSION

**Q1:** ZERO-SHOT HPO AS A STAND-ALONE PROBLEM

In Table 3 we report the final normalized regret achieved by the different zero-shot approaches for the first 20 hyper-parameters (Feurer et al. (2015)). Our method provides dataset-conditioned hyper-parameters that perform better than heuristics for small budgets[4]. The use of engineered meta-features to represent datasets for HPO solutions is not reliable, as the results achieved by NN-$\langle$MF1$\rangle$ and NN-$\langle$MF2$\rangle$ are no better than random. On the other hand, using the meta-features extracted from the dataset directly, NN-$\langle$D2V$\rangle$serves as a better approximation. Furthermore, random sampling from the restricted hyper-ellipsoid also outperforms the use of initialization strategies based on meta-features. We obtain the zero-shot hyper-parameters via Algorithm 2. The **D2V** meta-features are obtained via Algorithm 5.

Table 3: Results on several zero-shot HPO benchmarks. The numbers reported are the average normalized regret for 120 tasks on each meta-dataset evaluated as the average of a 5-fold cross-validation scheme. We report the best results in **bold** and underline the second best.

| Method | Zero-shot @5 | | | Zero-shot @20 | | |
|---|---|---|---|---|---|---|
| | Layout Md | Regularization Md | Optimization Md | Layout Md | Regularization Md | Optimization Md |
| Random | $13.988 \pm 2.629$ | $15.105 \pm 1.550$ | $12.836 \pm 1.781$ | $6.900 \pm 1.274$ | $8.909 \pm 0.440$ | $7.185 \pm 0.738$ |
| Average Rank | $13.103 \pm 2.679$ | $11.339 \pm 1.216$ | $11.843 \pm 1.647$ | $6.660 \pm 1.947$ | $\mathbf{5.362} \pm 0.772$ | $7.128 \pm 1.148$ |
| NN-MF1 | $13.544 \pm 1.999$ | $12.970 \pm 2.157$ | $12.300 \pm 1.722$ | $7.430 \pm 1.443$ | $7.542 \pm 0.809$ | $7.164 \pm 1.404$ |
| NN-MF2 | $14.350 \pm 2.181$ | $12.137 \pm 1.008$ | $12.717 \pm 2.228$ | $7.621 \pm 1.025$ | $6.788 \pm 1.122$ | $8.055 \pm 1.868$ |
| NN-D2V | $12.278 \pm 2.430$ | $13.831 \pm 3.556$ | $11.568 \pm 0.671$ | $6.526 \pm 2.215$ | $6.820 \pm 1.932$ | $6.695 \pm 0.942$ |
| Ellipsoid | $\underline{11.497} \pm 2.236$ | $\underline{11.114} \pm 1.433$ | $\mathbf{10.713} \pm 1.061$ | $\underline{5.713} \pm 1.776$ | $6.592 \pm 1.047$ | $\underline{6.665} \pm 1.154$ |
| GROSI | $\mathbf{10.945} \pm 1.602$ | $\mathbf{10.779} \pm 1.626$ | $\underline{11.298} \pm 1.086$ | $\mathbf{5.707} \pm 1.539$ | $\underline{5.903} \pm 1.655$ | $\mathbf{6.242} \pm 1.223$ |

---

[4]We depict the surrogate with the response model in F.2 as a plausibility check.

**Q2:** ZERO-SHOT HPO AS AN INITIALIZATION STRATEGY FOR SINGLE-TASK SEQUENTIAL HPO METHODS

We use the aforementioned initialization strategies to warm-start single task GP with a Matern 3/2 kernel and automatic relevance determination as the response model. The quality of our suggested hyper-parameters is reflected in the improved performance of the response model at the early stages compared to metafeature-based initialization and random search, Figure 1. The pseudo-code for sequential model-based optimization is provided by Algorithm 3.

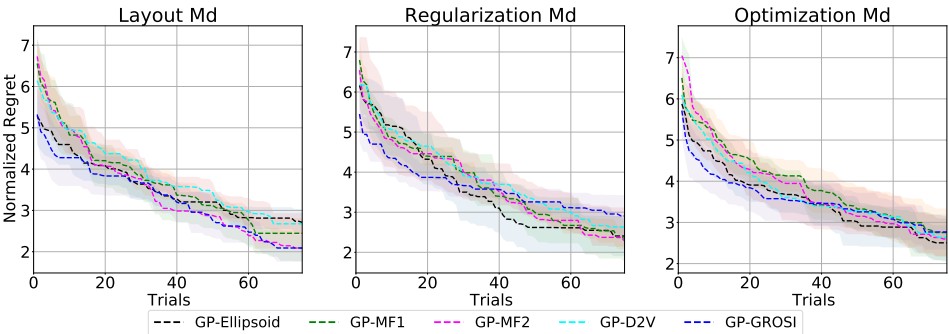

Figure 1: Average normalized regret for single-task sequential HPO methods, a Gaussian process, with different initialization strategies.

**Q3:** SEQUENTIAL GRAY-BOX FUNCTION OPTIMIZATION

The proposed universal response model provides useful hyper-parameters upfront without access to any observations of losses of the target responses. However, by iteratively refitting the model to the history of observations, the response prediction is improved, as depicted in Figure 5, and summarized in Table 4. We refit our algorithm by optimizing Equation 3, on the history of observed losses on the target dataset, Algorithm 4. We evaluate two policies for selecting the next hyper-parameter after refitting, (1) greedily selecting the hyper-parameter with the highest predicted response, GROSI(+1), and (2) selecting the next hyper-parameter randomly from the top 5 hyper-parameters with the highest predicted response, GROSI(+10), which achieved the best regret on average across the three meta-datasets, Appendix E.2. In contrast to the baselines that select hyper-parameters through an acquisition function that capitalizes on the uncertainty of the posterior samples, we incorporate uncertainty by selecting the next hyper-parameter from the top-k hyper-parameters uniformly at random and thus introduce a small trade-off between exploration and exploitation.

Furthermore, our method outperforms the state-of-the-art in transfer learning approaches for HPO in several cases while demonstrating in general competitive performance across all three meta-datasets, Table 4[5]. The baselines are warm-started with 20 randomly selected hyper-parameters (Feurer et al. (2015). For better readability, the uncertainty quantification can be found in Figure 4.

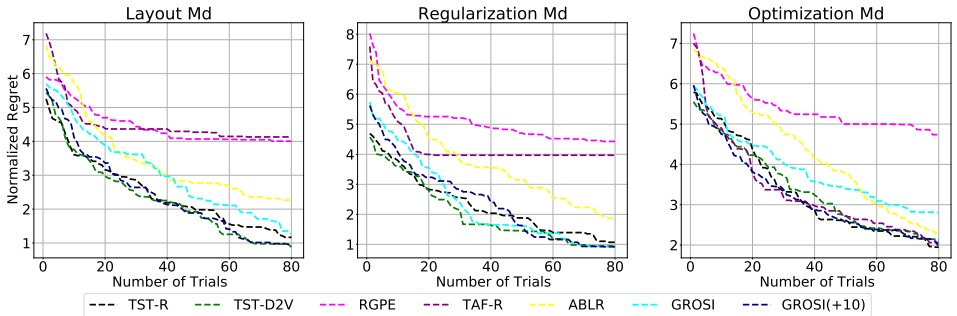

Figure 2: Average normalized regret for state-of-the-art transfer learning HPO methods.

---

[5]For better visualization, some baselines are removed from Figure 2, but are still reported in Table 4

Table 4: Normalized regret for state-of-the-art transfer learning HPO methods for up to 80 trials after initialization with 20 configurations.

| Method | @50 after initialization | | | @80 after initialization | | |
|---|---|---|---|---|---|---|
| | Layout Md | Regularization Md | Optimization Md | Layout Md | Regularization Md | Optimization Md |
| SMFO | $2.129 \pm 0.216$ | $\mathbf{1.498} \pm 0.572$ | $\mathbf{2.081} \pm 1.166$ | $1.498 \pm 0.480$ | $1.183 \pm 0.399$ | $\mathbf{1.822} \pm 0.873$ |
| TST-R | $1.982 \pm 0.329$ | $1.879 \pm 1.153$ | $2.611 \pm 0.852$ | $1.168 \pm 0.275$ | $1.070 \pm 0.989$ | $1.947 \pm 0.492$ |
| TST-D2V | $1.921 \pm 0.880$ | $\underline{1.444} \pm 0.698$ | $2.747 \pm 0.731$ | $\mathbf{0.877} \pm 0.501$ | $0.942 \pm 0.592$ | $2.126 \pm 1.037$ |
| RGPE | $4.069 \pm 0.515$ | $4.706 \pm 0.935$ | $5.000 \pm 0.572$ | $4.004 \pm 0.503$ | $4.429 \pm 0.742$ | $4.736 \pm 0.799$ |
| TAF-R | $4.270 \pm 0.805$ | $3.969 \pm 0.545$ | $2.714 \pm 1.013$ | $4.131 \pm 1.070$ | $3.969 \pm 0.545$ | $2.059 \pm 0.757$ |
| ABLR | $2.770 \pm 0.583$ | $3.140 \pm 1.654$ | $3.775 \pm 1.839$ | $2.267 \pm 0.549$ | $1.824 \pm 1.175$ | $2.299 \pm 0.799$ |
| MetaBO | $7.422 \pm 0.942$ | $7.145 \pm 1.222$ | $6.468 \pm 0.921$ | $7.422 \pm 0.942$ | $7.127 \pm 1.247$ | $6.131 \pm 1.036$ |
| GROSI | $2.313 \pm 0.828$ | $1.621 \pm 0.491$ | $3.397 \pm 0.915$ | $1.274 \pm 0.977$ | $\mathbf{0.906} \pm 0.258$ | $2.806 \pm 0.969$ |
| GROSI(+1) | $\mathbf{1.599} \pm 0.399$ | $1.637 \pm 0.752$ | $2.598 \pm 1.369$ | $1.118 \pm 0.635$ | $1.000 \pm 0.613$ | $\underline{1.871} \pm 1.208$ |
| GROSI(+10) | $\underline{1.887} \pm 0.546$ | $1.631 \pm 0.706$ | $\underline{2.557} \pm 1.035$ | $\underline{0.956} \pm 0.658$ | $\underline{0.922} \pm 0.308$ | $1.950 \pm 0.911$ |

## 5.5 ABLATION STUDY

We perform several ablation experiments to analyze the contribution of each objective to the overall performance. The results are detailed in Table 5. Treating zero-shot HPO as a simple regression model by optimizing Equation 3 alone is suboptimal and does not scale across all meta-datasets. We notice that adding the auxiliary dataset identification task, Equation 6 brings on significant improvement, similarly with the similarity driven regularization, Equation 7. This reinforces the notion that the responses of similar datasets behave similarly with regards to the hyper-parameters. Both losses help generate more expressive meta-features, the former more directly, by optimizing the inter- and intra-dataset similarities, and the latter indirectly by penalizing the difference in the predicted response.

Table 5: Final results for zero-shot HPO for different variations of our model optimized with different objectives. The numbers reported are the average normalized regret after 20 trials.

| Method | Layout Md | Regularization Md | Optimization Md |
|---|---|---|---|
| $\mathcal{O}$ | $7.263 \pm 1.309$ | $5.961 \pm 1.841$ | $7.625 \pm 0.546$ |
| $\mathcal{O} + \alpha\mathcal{R}$ | $7.121 \pm 1.133$ | $6.217 \pm 2.369$ | $6.488 \pm 0.609$ |
| $\mathcal{O} + \delta\mathcal{P}$ | $\underline{6.139} \pm 1.654$ | $6.105 \pm 1.765$ | $\underline{6.332} \pm 1.516$ |
| Pretrained Meta-feature Extractor $\phi$ | | | |
| $\mathcal{O} + \alpha\mathcal{R}$ | $6.698 \pm 1.099$ | $\underline{5.887} \pm 0.902$ | $7.152 \pm 1.287$ |
| $\mathcal{O} + \alpha\mathcal{R} + \delta\mathcal{P}$ | $6.516 \pm 0.911$ | $\mathbf{5.650} \pm 1.678$ | $9.925 \pm 7.130$ |
| GROSI | $\mathbf{5.707} \pm 1.539$ | $5.903 \pm 1.655$ | $\mathbf{6.242} \pm 1.223$ |

We also initialize the meta-feature extractor, $\phi$, by pretraining it independently, Algorithm 5. However, we notice that this leads to generally poor performance as the model arrives quickly at a local optimum. An artifact of the meta-dataset, we notice that pretraining GROSI for Regularization Md provides a small lift. A small sensitivity analysis can be found in Appendix F.1.

## 6 CONCLUSION

In this paper, we formulate HPO as a gray-box function optimization problem that incorporates an important domain of the response function, the dataset itself. We design a novel universal response model for zero-shot HPO that provides good initial hyper-parameters for unseen datasets in the absence of associated observations of hyper-parameters. We propose and optimize a novel multi-task objective to estimate the response while learning expressive dataset meta-features. We also reinforce the assumption that similar datasets behave similarly to hyper-parameters by introducing a novel similarity-driven regularization technique. As part of future work, we will investigate the impact of our approach within the reinforcement learning framework.

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

# A  DETAILED PROBLEM SETTING

By a **learning task** we denote a pair $(p, \ell)$ of an unknown distribution $p$ of pairs $(x, y) \in \mathbb{R}^{M+L}$, with $M, L \in \mathbb{N}$, and a loss $\ell : \mathbb{R}^L \times \mathbb{R}^L \to \mathbb{R}$. A function $\hat{y} : \mathbb{R}^M \to \mathbb{R}^L$ is called a **model for task** $p$ and

$$\ell(\hat{y}; p) := \mathbb{E}_{(x,y)\sim p}(\ell(y, \hat{y}(x)))$$

its (expected) **loss**.

Let $a$ be a **learning algorithm** that yields for every sample $D$ of pairs $(x, y)$ from a task $(p, \ell)$ and hyper-parameters $\lambda \in \mathbb{R}^P$ a model $\hat{y}$ for the task. We call

$$\ell(\lambda) := \ell(a(D, \lambda); p)$$

the loss (or the response) of hyper-parameters $\lambda$. We say validation loss for the loss estimated on fresh validation data.

**Sequential single-task hyper-parameter optimization problem.** Given an initial number $K$ of pairs $(\lambda_k, l_k)$ of hyper-parameters and their (validation) losses and a budget $B \in \mathbb{N}$ of trials, find sequentially $B$ many hyper-parameters $\lambda_{K+1}, \ldots, \lambda_{K+B}$, such that their smallest loss

$$\min_{k \in 1:K+B} \ell(\lambda_k)$$

is minimal among all such sequences. To compute the next guess $\lambda_{k+1}$, the hyper-parameters $\lambda_1, \ldots, \lambda_k$ tried so far and their (validation) losses $l_k := \ell(\lambda_k)$ can be used.

**Zero-shot cross-task hyper-parameter optimization problem.** Let $p^{\text{task}}$ be an unknown distribution of supervised learning tasks. Given a sample of triples $((p, \ell), \lambda, l)$ of learning tasks $(p, \ell)$, hyper-parameters $\lambda$ and their losses $l$, find for a fresh task $(p, \ell)$ and a budget $B \in \mathbb{N}$ — without any observations of losses of hyper-parameters on this task — a set $\{\lambda_1, \ldots, \lambda_B\}$ of hyper-parameters, such that their smallest loss $\min_{k \in 1:B} \ell(\lambda_k)$ is minimal among all such sets.

**Sequential cross-task hyper-parameter optimization problem.** Given both, (i) a sample of triples $((p, \ell), \lambda, l)$ of learning tasks $(p, \ell)$, hyper-parameters $\lambda$ and their losses $l$, and (ii) a fresh task $(p, \ell)$ and a budget $B \in \mathbb{N}$, find sequentially $B$ many hyper-parameters $\lambda_1, \ldots, \lambda_B$, such that their smallest loss $\min_{k \in 1:B} \ell(\lambda_k)$ is minimal among all such sequences. To compute the next guess $\lambda_{k+1}$, the hyper-parameters $\lambda_1, \ldots, \lambda_k$ tried so far and their (validation) losses $l_k := \ell(\lambda_k)$ as well as all data on other tasks can be used.

# B  THE META-FEATURE EXTRACTOR

The meta-feature extractor is a set-based function, and is represented as an extended derivation of the Kolmogorov-Arnold representation theorem (Krková (1992)), which states that a multi-variate function $\phi$ can be defined as an aggregation of univariate functions over single variables:

$$\phi(x_1, \ldots, x_M) \approx \sum_{j=0}^{2M} h_m \big( \sum_{m=1}^{M} g_{m,j}(x_m) \big) \tag{9}$$

It is important to note that $\phi$ is permutation invariant, i.e. unaffected by any permutation on the input, which allows us to obtain the same output for the same multi-variate data regardless of the order of input. As a simple variant of this formulation (Zaheer et al. (2017)), we can replace the set of functions $h_m$, with single function $h$, and $g_{m,j}$ with a function $g$. In this paper, we incorporate the meta-feature extractor as part of the response model, effectively learning a conditional response on the dataset meta-features directly such that the approximation is defined as $\hat{\ell}(\lambda^{(t)}, \phi^{(t)}; \beta)$, with $\phi^{(t)} = \phi(D^{(t)})$.

## C  META-DATASETS

### C.1  LAYOUT HYPER-PARAMETER

Below are some examples of the number of neurons per layer for networks with different **layout** hyper-parameters given 4 neurons and 5 layers:

- Layout □: [4,4,4,4,4]
- Layout ◁: [4,8,16,32,64]
- Layout ▷: [64,32,16,8,4]
- Layout ◇: [4,8,16,8,4]
- Layout △: [16,8,4,8,16]

The search space consists of all possible combinations of the hyper-parameters. After removing redundant configurations, e.g. △ layout with 1 layer is similar to a □ layout with 1 layer, the resulting meta-datasets have $256$, $288$, and $324$ unique configurations respectively.

### C.2  HYPER-PARAMETER ENCODING

Below is description of the encodings applied to our hyper-parameters. We also like to note that the scalar values are normalized between $(0, 1)$.

Table 6: Encoding of the different hyper-parameters used in the meta-dataset.

| Hyper-parameter | Encoding |
|---|---|
| Activation | One-hot encoding |
| Neurons | Scalar |
| Layers | Scalar |
| Layout | One-hot encoding |
| Dropout | Scalar |
| Normalization | Scalar |
| Optimizer | One-hot encoding |

### C.3  THE UCI DATASETS

Table 7 is an overview of the UCI datasets used to generate the meta-datasets.

Table 7: Summary of the 120 UCI datasets used to generate the meta-datasets.

| UCI Dataset | # Instances | # Features | # Classes | UCI Dataset | # Instances | # Features | # Classes |
|---|---|---|---|---|---|---|---|
| molec-biol-splice | 2393 | 60 | 3 | adult | 32561 | 14 | 2 |
| twonorm | 5550 | 20 | 2 | annealing | 798 | 31 | 5 |
| plant-texture | 1199 | 64 | 100 | molec-biol-promoter | 80 | 57 | 2 |
| ringnorm | 5550 | 20 | 2 | contrac | 1105 | 9 | 3 |
| spect | 79 | 22 | 2 | statlog-landsat | 4435 | 36 | 6 |
| energy-y2 | 576 | 8 | 3 | conn-bench-sonar-mines-rocks | 156 | 60 | 2 |
| steel-plates | 1456 | 27 | 7 | musk-2 | 4949 | 166 | 2 |
| vertebral-column-3clases | 233 | 6 | 3 | balloons | 12 | 4 | 2 |
| chess-krvk | 21042 | 6 | 18 | abalone | 3133 | 8 | 3 |
| statlog-shuttle | 43500 | 9 | 7 | statlog-vehicle | 635 | 18 | 4 |
| breast-cancer-wisc | 524 | 9 | 2 | page-blocks | 4105 | 10 | 5 |
| semeion | 1195 | 256 | 10 | heart-hungarian | 221 | 12 | 2 |
| connect-4 | 50668 | 42 | 2 | ionosphere | 263 | 33 | 2 |
| monks-3 | 122 | 6 | 2 | synthetic-control | 450 | 60 | 6 |
| wall-following | 4092 | 24 | 4 | plant-shape | 1200 | 64 | 100 |
| vertebral-column-2clases | 233 | 6 | 2 | pittsburg-bridges-MATERIAL | 80 | 7 | 3 |
| planning | 137 | 12 | 2 | breast-cancer-wisc-diag | 427 | 30 | 2 |
| cardiotocography-3clases | 1595 | 21 | 3 | spectf | 80 | 44 | 2 |
| plant-margin | 1200 | 64 | 100 | bank | 3391 | 16 | 2 |
| nursery | 9720 | 8 | 5 | pendigits | 7494 | 16 | 10 |
| titanic | 1651 | 3 | 2 | teaching | 113 | 5 | 3 |
| energy-y1 | 576 | 8 | 3 | mushroom | 6093 | 21 | 2 |
| monks-1 | 124 | 6 | 2 | optical | 3823 | 62 | 10 |
| arrhythmia | 339 | 262 | 13 | primary-tumor | 248 | 17 | 15 |
| breast-tissue | 80 | 9 | 6 | conn-bench-vowel-deterding | 528 | 11 | 11 |
| statlog-australian-credit | 518 | 14 | 2 | soybean | 307 | 35 | 18 |
| tic-tac-toe | 719 | 9 | 2 | oocytes_merluccius_states_2f | 767 | 25 | 3 |
| lymphography | 111 | 18 | 4 | chess-krvkp | 2397 | 36 | 2 |
| monks-2 | 169 | 6 | 2 | audiology-std | 171 | 59 | 18 |
| waveform | 3750 | 21 | 3 | image-segmentation | 210 | 18 | 7 |
| fertility | 75 | 9 | 2 | led-display | 750 | 7 | 10 |
| lenses | 18 | 4 | 3 | heart-va | 150 | 12 | 5 |
| wine-quality-red | 1199 | 11 | 6 | pittsburg-bridges-SPAN | 69 | 7 | 3 |
| parkinsons | 146 | 22 | 2 | oocytes_trisopterus_nucleus_2f | 684 | 25 | 2 |
| wine-quality-white | 3674 | 11 | 7 | statlog-german-credit | 750 | 24 | 2 |
| pima | 576 | 8 | 2 | acute-inflammation | 90 | 6 | 2 |
| pittsburg-bridges-T-OR-D | 77 | 7 | 2 | car | 1296 | 6 | 4 |
| low-res-spect | 398 | 100 | 9 | horse-colic | 300 | 25 | 2 |
| musk-1 | 357 | 166 | 2 | heart-switzerland | 92 | 12 | 5 |
| pittsburg-bridges-REL-L | 77 | 7 | 3 | oocytes_trisopterus_states_5b | 684 | 32 | 3 |
| breast-cancer | 215 | 9 | 2 | congressional-voting | 326 | 16 | 2 |
| spambase | 3451 | 57 | 2 | acute-nephritis | 90 | 6 | 2 |
| iris | 113 | 4 | 3 | credit-approval | 518 | 15 | 2 |
| thyroid | 3772 | 21 | 3 | hill-valley | 606 | 100 | 2 |
| mammographic | 721 | 5 | 2 | oocytes_merluccius_nucleus_4d | 767 | 41 | 2 |
| ilpd-indian-liver | 437 | 9 | 2 | seeds | 158 | 7 | 3 |
| blood | 561 | 4 | 2 | ozone | 1902 | 72 | 2 |
| waveform-noise | 3750 | 40 | 3 | magic | 14265 | 10 | 2 |
| statlog-heart | 203 | 13 | 2 | statlog-image | 1733 | 18 | 7 |
| pittsburg-bridges-TYPE | 79 | 7 | 6 | cylinder-bands | 384 | 35 | 2 |
| echocardiogram | 98 | 10 | 2 | lung-cancer | 24 | 56 | 3 |
| flags | 146 | 28 | 8 | dermatology | 275 | 34 | 6 |
| letter | 15000 | 16 | 26 | cardiotocography-10clases | 1595 | 21 | 10 |
| zoo | 76 | 16 | 7 | heart-cleveland | 227 | 13 | 5 |
| ecoli | 252 | 7 | 8 | haberman-survival | 230 | 3 | 2 |
| yeast | 1113 | 8 | 10 | balance-scale | 469 | 4 | 3 |
| hayes-roth | 132 | 3 | 3 | wine | 134 | 13 | 3 |
| libras | 270 | 90 | 15 | miniboone | 97548 | 50 | 2 |
| breast-cancer-wisc-prog | 149 | 33 | 2 | hepatitis | 116 | 19 | 2 |
| glass | 161 | 9 | 6 | post-operative | 68 | 8 | 3 |

## D    Algorithms

We define $p_{\mathcal{T}}$ as the task distribution that represents pairs of datasets and hyper-parameters, i.e. $\mathcal{T}^{(t,k)} = (D^{(t)}, \lambda_k^{(t)}) \in T \times \Lambda$, and $p_D$ be the distribution of the datasets as defined in Section 4.2. Algorithm 1 provides the overall optimization framework for GROSI, our approach.

---

**Algorithm 1 Learn_GROSI**$(D)$

---

1: **Require:** $p_D$: distribution over datasets, $p_T$: distribution over tasks
2: **Require:** $lr_{\text{inner}}$, $lr_{\text{outer}}$: learning rates
3: Randomly initialize $\beta \in \mathcal{B}$, the parameters of our response model $\hat{l}$
4: **while** not done **do**
5:      Set $\beta' \leftarrow \beta$
6:      Sample $(D^{(t)}, \lambda_k^{(t)}) = \mathcal{T}^{(t,k)} \sim p_{\mathcal{T}}$
7:      **for** $v$ steps
8:          sample $D^{(q)} \sim p_{D \setminus (D^{(t)}}$
9:          Evaluate gradients $\mathcal{G} \leftarrow \nabla_\beta (\mathcal{O} + \alpha\,\mathcal{R} + \delta\,\mathcal{P})$
10:          Compute adapted parameters with stochastic gradient descent: $\beta' \leftarrow \beta' - lr_{\text{inner}}\mathcal{G}$
11:      Update $\beta \leftarrow \beta - lr_{\text{outer}} (\beta - \beta')$
12: **return** $\beta$

---

After optimizing our objective via Algorithm D we apply Algorithm 2 to observe the results presented in Tables 3 and 5.

---

**Algorithm 2** Zero-shot HPO

---

1: **Require:** target dataset $D^{(t)}$ ; response model $\hat{l}$; desired zero-shot hyper-parameters $K$
2: $\mathcal{H} \leftarrow \arg\min_{\lambda \in \Lambda}^{K} \hat{l}\left(D^{(t)}, \lambda\right)$
3: **return** $\mathcal{H}$

---

For sequential model-based optimization, a surrogate $\hat{l}$ is fitted to the observed responses of the unknown function. Several initialization strategies exist to expedite the transfer of information across tasks, Section 5.4. In Algorithm 3, we present the generic pseudo-code for SMBO, that requires an acquisition function, $a$, to sample the next iterate from the domain.

---

**Algorithm 3** Sequential Model-based Optimization Warm-start

---

1: **Require:** target dataset $D^{(t)}$ ; response model $\hat{l}$; desired zero-shot hyper-parameters $K$, number of trials $I$, acquisition function $a$
2: Get initial hyper-parameters $\mathcal{H}_0 \leftarrow$ Zero-shot HPO
3: $\lambda^{\min} \leftarrow \arg\min_{\lambda \in \mathcal{H}_0} \left(l(D^{(t)}, \lambda)\right)$
4: **for** $i = 1 \ldots I$
5:      fit $\hat{l}_i$ to $\mathcal{H}_{i-1}$
6:      $\lambda \leftarrow \arg\max_{\lambda \in \Lambda} a\left(\hat{l}(D^{(t)}, \lambda)\right)$
7:      $\mathcal{H}_i \leftarrow \mathcal{H}_{i-1} \bigcup \{\lambda\}$
8:      **if** $l\left(D^{(t)}, \lambda\right) < l\left(D^{(t)}, \lambda^{\min}\right)$
9:          $\lambda^{\min} \leftarrow \lambda$
10: **return** $\lambda^{\min}$

---

In Section 5.4, we propose to initialize our response model on the target dataset, then iterativly tune it to that particular dataset. Initially, we select top $K$ configurations based on Algorithm 2, our zero-shot approach. Then, via Algorithm 4, we sample uniformly at random from the top $X$ ranking configurations. If $X = 1$, then this represents the greedy policy.

Meta-feature learning from datasets with varying schema was initially proposed in (Jomaa et al. (2019)). For our approach, we introduce a set-based meta-feature extractor module to handle datasets

---

**Algorithm 4** Learn_GROSI(+X)

---

1: **Require:** target dataset $D^{(t)}$ ; response model $\hat{l}$; desired zero-shot hyper-parameters $K$, number of trials $I$, Number of top configurations to choose from, $X$
2: Get initial hyper-parameters $\mathcal{H}_0 \leftarrow$ Zero-shot HPO
3: $\lambda^{\min} \leftarrow \arg\min_{\lambda \in \mathcal{H}_0} \left( l(D^{(t)}, \lambda) \right)$
4: **for** $i = 1 \ldots I$
5:      fit $\hat{l}_i$ to $\mathcal{H}_{i-1}$ by optimizing Equation 3
6:      $\lambda \sim \text{Uniform} \left( \arg\min^X_{\lambda \in \Lambda \setminus \mathcal{H}_{i-1}} \hat{l} \left( D^{(t)}, \lambda \right) \right)$
7:      $\mathcal{H}_i \leftarrow \mathcal{H}_{i-1} \bigcup \{\lambda\}$
8:      **if** $l \left( D^{(t)}, \lambda \right) < l \left( D^{(t)}, \lambda^{\min} \right)$
9:          $\lambda^{\min} \leftarrow \lambda$
10: **return** $\lambda^{\min}$

---

of varying schema as well, however, we optimize Equation 8 and use the dataset identification task as an auxiliary objective. However, to pre-train the meta-feature extractor for the Ablation study, Section 5.5, as well as in order to extract meta-features for the NN-D2V, and TST-D2V, we follow Algorithm 5, with $p_{D+}$ as the distribution of similar datasets, $p_{D+} = \{(D^{(t)}, D^{(q)}, s) \sim p_D \mid s = 1\}$, and $p_{D-}$ as the distribution of dissimilar datasets, $p_{D-} = \{(D^{(t)}, D^{(q)}, s) \sim p_D \mid s = 0\}$. Similar datasets are defined as multi-fidelity subsets (batches) of each dataset.

---

**Algorithm 5** Standalone Meta-feature Learning

---

1: **Require:** $p_{D+}$: distribution over similar datasets, $p_{D-}$ distribution over dissimilar datasets
2: **Require:** $lr_\phi$ learning rate
3: Randomly initialize $\beta \in \mathcal{B}$, the parameters of the meta-feature extractor $\phi$
4: **while** not done **do**
5:      Sample $(D^{(t)}, D^{(q)}, 1) \sim p_{D+}$ and $(D^{(t)}, D^{(r)}, 0) \sim p_{D-}$ (Both samples share $D^{(t)}$)
6:      Evaluate gradients $\mathcal{G} \leftarrow \nabla_\beta (\mathcal{P})$
7:      Compute adapted parameters with stochastic gradient descent: $\beta' \leftarrow \beta' - lr_\phi \mathcal{G}$
8:      Update $\beta \leftarrow \beta - lr_\phi (\beta - \beta')$
9: **return** $\beta$

---

# E  EXPERIMENTAL DETAILS

## E.1  NETWORK ARCHITECTURE

Our model architecture is divided into two modules, $\hat{l} := \phi \circ \psi$, the meta-feature extractor $\phi$, and the regression head $\psi$. The meta-feature extractor $\phi : \mathbb{R}^2 \to \mathbb{R}^{K_h}$ is composed of three functions, Equation 4, namely $f$, $g$ and $h$. The regression head is also composed of two functions, i.e. $\psi : \psi_1 \circ \psi_2$. We define by $\psi_1 : \mathbb{R}^{K_h} \times \Lambda \to \mathbb{R}^{K_{\psi_1}}$ as the function that takes as input the meta-feature/hyper-parameter pair, and by $\psi_2 : \mathbb{R}^{K_{\psi_1}} \to \mathbb{R}$ the function that approximates the response. Finally, let **Dense(n)** define one fully connected layer with $n$ neurons, and **ResidualBlock(n,m)** be $m \times$ Dense(n) with residual connections (Zagoruyko & Komodakis (2016)).

To select a single universal response model, we evaluate the validation performance on the three network architectures described in Table 8. We select the architecture that has the best average performance between the three across the three meta-datasets, Table 9, which turns out to be Architecture 3. The architectures assign a different number of trainable variables for the meta-feature extractor and the coupled regression head.

Table 8: The network architecture optimized for every meta-dataset.

| Functions | Architecture 1 |
|---|---|
| $f$ | Dense(32);6×ResidualBlock(3,32);Dense(32) |
| $g$ | Layout: $\triangleright$ with 3 layers and 16 Neurons |
| $h$ | Dense(16);3×ResidualBlock(3,16);Dense(16) |
| $\psi_1$ | Layout: $\triangleright$ with 4 layers and 4 Neurons |
| $\psi_2$ | Layout: $\triangleright$ with 4 layers and 4 Neurons |
| Functions | Architecture 2 |
| $f$ | Dense(16);6×ResidualBlock(3,16);Dense(16) |
| $g$ | Layout: $\triangleright$ with 3 layers and 16 Neurons |
| $h$ | Dense(32);3×ResidualBlock(3,32);Dense(32) |
| $\psi_1$ | Layout: $\triangleright$ with 4 layers and 4 Neurons |
| $\psi_2$ | Layout: $\triangleright$ with 4 layers and 16 Neurons |
| Functions | Architecture 3 |
| $f$ | Dense(16);6×ResidualBlock(3,16);Dense(16) |
| $g$ | Layout: $\triangleright$ with 3 layers and 16 Neurons |
| $h$ | Dense(16);3×ResidualBlock(3,16);Dense(16) |
| $\psi_1$ | Layout: $\triangleright$ with 4 layers and 4 Neurons |
| $\psi_2$ | Layout: $\triangleright$ with 4 layers and 16 Neurons |

Table 9: Final results of each model optimized on the different meta-datasets. The numbers reported are the average normalized regret after 50 trials on **held-out validation** sets for the zero-shot task.

| Architecture | Layout Md | Regularization Md | Optimization Md | Average Score |
|---|---|---|---|---|
| Architecture 1 | $0.942 \pm 0.2074$ | $1.697 \pm 0.2897$ | $2.935 \pm 1.4562$ | $1.858 \pm 0.821$ |
| Architecture 2 | $0.973 \pm 0.5338$ | $1.849 \pm 0.5397$ | $3.215 \pm 1.3578$ | $2.012 \pm 0.922$ |
| Architecture 3 | $1.041 \pm 0.4134$ | $1.715 \pm 0.6354$ | $2.579 \pm 0.9865$ | $\mathbf{1.779} \pm 0.629$ |

## E.2  POLICY FOR SEQUENTIAL OPTIMIZATION

We propose GROSI as a zero-shot HPO solution. However, to emphasize the ability of our surrogate model to quickly adapt to new target datasets, we extend it into a sequential optimization approach. Starting with the proposed zero-shot configurations, we fine tune our model via Algorithm 4. We select the $X = 10$ based on the best average performance observed on the held-out validation sets, Table10.

Table 10: Final results for different variants of our sequential model optimization policy. The numbers reported are the average normalized regret after 80 trials on **held-out validation** after initialization with the exact same 20 configurations suggested by the our zero-shot approach.

| Sequential Policy | Layout Md | Regularization Md | Optimization Md | Average Score |
|---|---|---|---|---|
| GROSI(+1) | 0.653±0.540 | 1.793±1.057 | 2.835±0.663 | 1.760±0.891 |
| GROSI(+3) | 0.204±0.265 | 1.845±1.252 | 2.738±1.038 | 1.596±1.049 |
| GROSI(+5) | 0.403±0.402 | 2.027±1.171 | 2.480±0.967 | 1.637±0.892 |
| GROSI(+7) | 0.675±0.416 | 1.576±0.845 | 2.476±1.129 | 1.576±0.735 |
| GROSI(+10) | 0.550±0.498 | 1.727±1.075 | 1.996±1.191 | **1.424**±0.628 |

## F    ADDITIONAL EXPERIMENTAL RESULTS

### F.1    HYPER-PARAMETER SENSITIVITY ANALYSIS

We optimize our response model by minimizing Equation 8, which includes the dataset identification task, Equation 6, and the similarity-driven regularization task, Equation 7, with auxiliary weights $\delta$ and $\alpha$ assigned to both respectively. We report below the performance of our universal response model for different auxiliary weights. The results confirm the importance of emphasizing the auxiliary dataset identification task in conjunction with the similarity-driven regularization loss, which reinforces the intuition that similar datasets behave similarly to the hyper-parameter response. The reported results throughout the paper are based on $\delta = 1$ and $\alpha = 0.5$. .

.

Table 11: Final results of our universal response model optimized with different auxiliary weights. The numbers reported are the average normalized regret after 20 trials.

| Method | Layout Md | Regularization Md | Optimization Md |
|---|---|---|---|
| $\delta = 0.5; \alpha = 0.1$ | $\underline{6.016} \pm 1.196$ | $\mathbf{5.358} \pm 1.676$ | $6.744 \pm 0.468$ |
| $\delta = 0.5; \alpha = 0.5$ | $6.973 \pm 2.525$ | $6.483 \pm 1.614$ | $\underline{6.401} \pm 0.325$ |
| GROSI(Ours):$\delta = 1; \alpha = 0.5$ | $\mathbf{5.707} \pm 1.539$ | $\underline{5.903} \pm 1.655$ | $\mathbf{6.242} \pm 1.223$ |

### F.2    ADDITIONAL RESULTS

#### Q1: ZERO-SHOT HPO AS A STAND-ALONE PROBLEM

As a plausibility argument for the usefulness of our zero-shot strategy, we depict in Figure 3 the top 10 suggested hyper-parameters by our approach, as well as two initialization strategies on the actual response surface. Our picks can be seen colocated near the different optima in the search space whereas hyper-parameters of other strategies are dispersed.

#### Q3: SEQUENTIAL GRAY-BOX FUNCTION OPTIMIZATION

We refit the universal response model to the observations of the response on the target dataset by optimizing Equation 3. We depict the improvement achieved over the zero-shot approach in Figure 5.

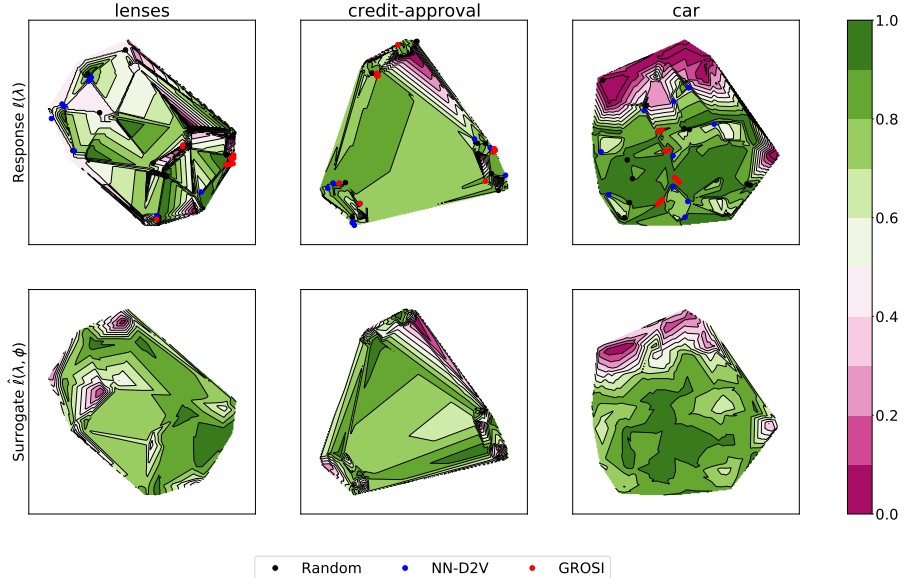

Figure 3: Top 10 hyper-parameters suggested by our approach for test datasets from the three different meta-datasets. We reduce the dimensionality of each search space into a 2D representation via TSNE (Liu et al. (2016)). The first row represents the actual response surface. We notice that the true response and the predicted response are similar, and the location of the predicted minima, in green, overlaps with the minima of the actual response.

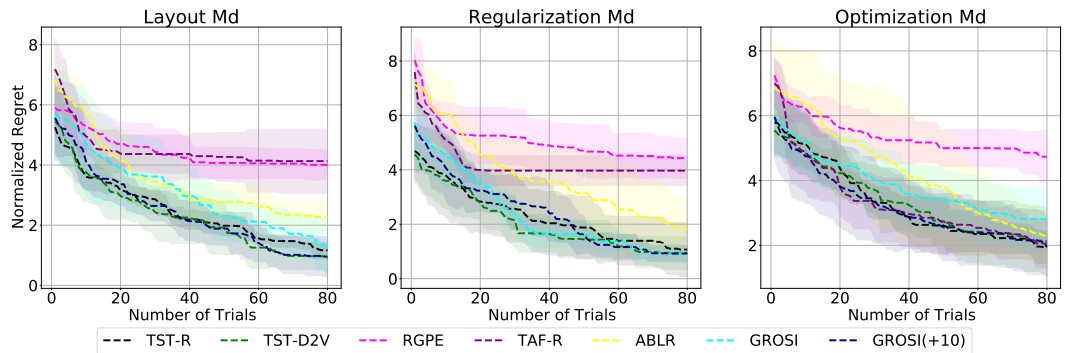

Figure 4: Average normalized regret for state-of-the-art transfer learning HPO methods, with uncertainty quantification.

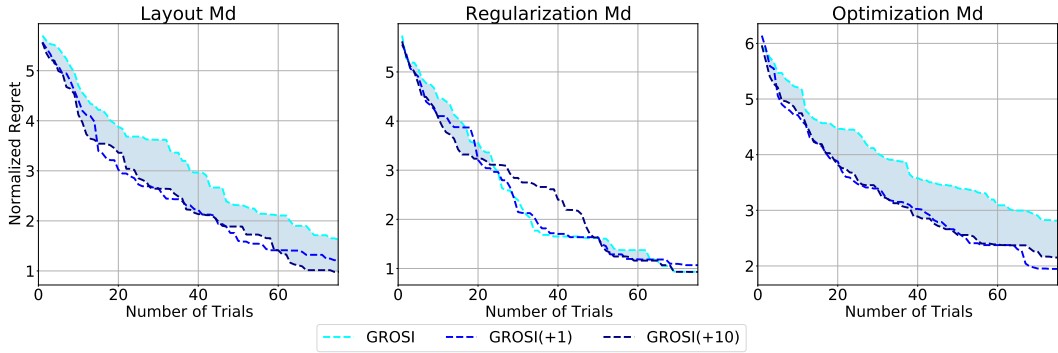

Figure 5: : Average normalized regret for GROSI, our zero-shot approach, and the refitted response models GROSI(+1) and GROSI(+10). We shade in light blue the improvement over zero-shot performance.

