# OpenReview forum: "Zero-shot Transfer Learning for Gray-box Hyper-parameter Optimization"
_ICLR.cc/2021/Conference — Reject_

### Official Review · AnonReviewer1 · 2020-10-20
**This work applies meta-learning to zero-shot hyper-parameter optimization. No target response is required, and meta-features are extracted for the hyper-parameter estimation of a given dataset.**

**Rating:** 6
**Confidence:** 3

**Review:**

Pros:

This work proposes a solution for the zero-shot hyper-parameter optimization problem without access to observations of losses of the target response.

The proposed method is parallelizable and thus can reduce the required wall-clock time vastly.

Three new meta-datasets with different search spaces are proposed and served as a benchmark.

A battery of experiments and the completions comparison against state-of-the-art HPO solutions are conducted to demonstrate the performance of the proposed approach.


Cons:

This work may be treated as applying meta-learning to hyper-parameter optimization. That is, a model is introduced to learn how to learn a group of optimal parameters based on the given meta-features. The acquisition function, meta-features extraction, meta-learning optimization method, the dataset similarity loss, etc., are all borrowed from prior works.

The meta-features of datasets are directly computed by Dataset2Vec [1]. This paper points out that prior methods rely on engineered meta-features, but actually [1] can be applied to improve them as well. I suggest the authors discuss the situation where prior methods and [1] are combined to reveal the advantage of the proposed method.


[1] Jomaa, Hadi S., Lars Schmidt-Thieme, and Josif Grabocka. "Dataset2vec: Learning dataset meta-features." arXiv preprint arXiv:1905.11063 (2019).

---

> ### Author Response · Authors · 2020-11-14
> **Rebuttal to AnonReviewer1**
>
> **Comment**: This work may be treated as applying meta-learning to hyper-parameter optimization. That is, a model is introduced to learn how to learn a group of optimal parameters based on the given meta-features. The acquisition function, meta-features extraction, meta-learning optimization method, the dataset similarity loss, etc., are all borrowed from prior works.
>
> *Rebuttal*: Sure, as almost all papers we use a lot of existing methods. But our contribution is not just a new combination of pre-existing things, but we are the first to present a neural-network based surrogate function that regresses directly from the dataset to the response, without the use of pre-estimated (engineered or learnt) meta-features. We also design a novel similarity-driven regularization term, Equation 4.
>
> **Comment**: The meta-features of datasets are directly computed by Dataset2Vec [1]. This paper points out that prior methods rely on engineered meta-features, but actually [1] can be applied to improve them as well. I suggest the authors discuss the situation where prior methods and [1] are combined to reveal the advantage of the proposed method.
>
> *Rebuttal*: We borrow from Dataset2Vec the dataset identification task as an auxiliary task to improve regression on the response surface. Based on your suggestion, we present the algorithm for pretraining the meta-feature extractor via Algorithm 5.
>
> We use the learnt meta-features with the TST baseline, which we denote by TST-D2V. In the original paper, TST-R, performed much better than its counterpart TST-M, which relies on engineered meta-features. However, we witness in Figure 2 that TST-D2V is very competitive with TST-R, although not as good as our approach. On the other hand, we learn our meta-features *end-to-end* by optimizing Equation 8, page 5.
>
> We also present in the Ablation Study, Section 5.1, the effect of pretraining the meta-feature extractor and then optimizing our proposed objective.

---

### Official Review · AnonReviewer3 · 2020-10-22
**Interesting and efficient transfer learning HPO approach with some potential for improvement w.r.t clarity of model details, robustness of approach and relation to RL based HPO**

**Rating:** 7
**Confidence:** 3

**Review:**

The authors propose a novel approach to zero-shot transfer learning for hyper parameter optimisation (HPO). In contrast to standard HPO approaches, where HPs are found for each task by random/grid search or Bayesian Global Optimisation or sequential model based optimisation, the proposed approach learns a response model defined over the combined domain of datasets and hyper-parameters such that for a new task, for which training data and labels and some ranges of HPs are given, HPO can be done by querying the response model without having to evaluate the black-box function of the validation loss as a standard HPO would do.

I think the manuscript is very well written, the motivation is sound, the related work section includes many relevant and well summarized examples of similar approaches, the notation and the presentation of the proposed approach are intuitiv and concise.

One thing that could probably be improved a bit is the coherence of the text, I mean, I had to jump a lot between the Appendix and the paper to understand some necessary details. For instance the core of the approach in section 4, and eqs 3-5 are very difficult to understand without the information referenced in the last paragraph of that section. I wonder if it would be possible to mention some of the relevant information, e.g. how eq. 5 is optimized or how the dataset2vec features are learned, right in section 4.

The experimental evaluation appears solid and comprehensive. I like how the authors investigated their approach in three different settings. Compared with a large number of alternative approaches the proposed approach achieves improvements in all HPO settings, ‘cold-start’, ‘warm-start’ and continuous learning of the response model. One question I had when thinking about the experiments is that the proposed approach relies on a number of parameters to be chosen well, in particular the HPs that determine the computation of the meta features and the response model. It appears that the response model used some basic settings that were not optimised for the task, but I still wonder how sensitive the learning of the response model is to the choice of hyper parameters. That includes the 'generalisation performance' of the dataset2vec model and all other ingredients of the response model, apart from the validation loss of the black/grey-box ML model.

When comparing the proposed approach with existing approaches, it seems that at least some of the aspects that make this work so well should also be captured in other approaches that use some dataset features in a transfer learning HPO setting. Some of them are mentioned in the related work section (Perrone et al, Volpp et al, Feurer et al, …) but especially the experimental setting of SEQUENTIAL GRAY-BOX FUNCTION OPTIMIZATION looks very similar to reinforcement learning approaches to HPO which include dataset features in their policy (e.g. https://arxiv.org/abs/1906.11527). Given these similarities and the strong empirical results it would be great to get a better understanding of which elements that are different to existing work contribute most. The ablation study in Appendix F.2 is great, and could be discussed in the main text especially in relation to other work.

To summarise I think the experimental validation suggests that this approach has potential to improve the state of the art in HPO. Some relevant details of the method are not entirely clear, or at least could be better described in the main text. Also in the experimental evaluation it could made clearer that the results are not sensitive to the tuning of the, pretty complex, HPs of the response model itself.

---

> ### Author Response · Authors · 2020-11-14
> **Rebuttal to AnonReviewer3**
>
> **Comment**: It is suggested to re-organize the description of the proposed method. In the current version of the manuscript, many essential details are put in the appendix, such as the optimization of Eq. 5.
>
> *Rebuttal*: Based on your suggestion, we have reworked the problem setting section to be more self-contained.
>
> **Comment**:  ... but I still wonder how sensitive the learning of the response model is to the choice of hyper parameters. That includes the 'generalisation performance' of the dataset2vec model and all other ingredients of the response model,
>
> *Rebuttal*: In the updated draft, we report the results of a single model used for all three meta-datasets that was selected based on the best average normalized regret at 50 trials on the held-out validation sets, Section 4 page 5. The detailed scores are reported in Appendix E.
>
> We also provide a small scale sensitivity analysis on the coefficients of the auxiliary losses, Equation 4 and Equation 9, Appendix F.1.
>
> **Comment**: Given these similarities and the strong empirical results it would be great to get a better understanding of which elements that are different to existing work contribute most.
>
> *Rebuttal*: We agree that there are certain similarities between our approach and HYP-RL. However, the poor performance of standard Q-Learning in that paper makes it a poor baseline. Instead, we use the state-of-the-art in RL for black box optimization, MetaBO as a baseline, that also does not prove to be competitive.  But we definitely want to combine our method with RL in future work, as we point out in Section 6.
>
> **Comment**: The ablation study in Appendix F.2 is great, and could be discussed in the main text especially in relation to other work.
>
> Rebuttal: Thank you for your feedback. We have moved the ablation study to the main text as Section 5.5 and updated the results based on the new architecture used for all meta-datasets.

---

### Official Review · AnonReviewer2 · 2020-10-28
**This paper proposed a regularized regression using meta-feature embeddings of certain datasets as regularization for standard HBO.**

**Rating:** 6
**Confidence:** 3

**Review:**

In this papar, the authors formulated a new objective function for HBO, which included an additional regularization term based on the dataset similarity. The authors used the distance between the meta-features of selected datasets to measure this dataset similarity and assumpted that similar datasets should have similar hyper-paprameters.  The experiments are complete and demonstrate the advantages of using this new optimization formulation.

I have three major concerns as following.

1. In Appendix B.1, it mentioned that using the similarity regularization defined in Equation 4. alone is not sufficient to measure the dataset similarity, so an additional similarity metric Equation 9. is used. What is the impact of directly replacing the distance metric in Equation 4. with Equation 9?

2. Since the similarity regularization is computed using K selected datasets, and "for each meta-dataset the architecture of the models are selected based on their performance on their validation datasets", how to ensure that the proposed universal model trained on these K datasets can be generalized to other datasets.

3. It is suggested to re-organize the description of the proposed method. In the current version of the manuscript, many essential details are put in the appendix, such as the optimization of Eq. 5.

---

> ### Author Response · Authors · 2020-11-14
> **Rebuttal to AnonReviewer2**
>
> **Comment**: In Appendix B.1, it is mentioned that using the similarity regularization defined in Equation 4. alone is not sufficient to measure the dataset similarity, so an additional similarity metric Equation 9. is used. What is the impact of directly replacing the distance metric in Equation 4. with Equation 9?
>
> *Rebuttal*: We show this in our ablation study, Section 5.5, that Equation 4, the novel regularization loss which forces the response of similar datasets to be similar, and Equation 9, which represents the auxiliary dataset identification task, both improve the performance of our model when paired with the regression task, Equation 3. We notice that our approach works best when all are combined.
>
>
> **Comment**: Since the similarity regularization is computed using K selected datasets, and "for each meta-dataset the architecture of the models are selected based on their performance on their validation datasets", how to ensure that the proposed universal model trained on these K datasets can be generalized to other datasets.
>
> *Rebuttal*: We initially did a small scale search for the best architecture for every meta-dataset based on the best validation score on the respective held-out validation set. However, based on your suggestion, we now have selected the best architecture based on the average best validation score for each meta-dataset. We provide the architectures as well as the validation scores on the held out datasets in Appendix E.
>
> **Comment**: It is suggested to re-organize the description of the proposed method
>
> *Rebuttal*: We have used the extra page to improve the readability of the paper and make it more self-contained.

---

### Official Review · AnonReviewer4 · 2020-10-29
**Official Blind Review#4**

**Rating:** 6
**Confidence:** 3

**Review:**

Summary:
The authors propose a new zero-shot hyper-parameter optimization method based-on the meta-learning framework. The proposed method incorporates two ideas from the meta-learning framework namely the task similarity based on the meta-features and the dataset identification. The former idea is used to achieve the requirement that responses to similar data sets should be similar.The latter idea has the role of preventing data of dissimilar tasks from being embedded in close proximity in the meta-feature space.

In this paper, the authors construct a new meta-dataset to evaluate the proposed method by compare it with various HPO methods (both zero-shot approach and sequential model based approach).

While the performances of various HPO tasks certainly seems to have improved, there are some concerns.

- How can you justify the proposed approach  which measures domain similarity in terms of distances between meta-features  (some kind of point estimation), rather than inter-distributional distances? My concern is that the information may be considerably dropped than measuring similarity by inter-distributional distance.

- The algorithm in the appendix doesn't have a section on learning meta features, how (and when) are these updated in the algorithm?

- Appendix B.1 defines a regularization term for dataset identification.

  -- Although the range of \hat{s} is defined as {0, 1}, in (8) \hat{s} appears to be a continuous value. Are you using some kind of threshold to make it binary?

  -- The authors argue that  "Without loss of generality, we use the Euclidean distance to measure the similarity between the extracted meta features, ... and \gamma = 1". What do you mean by "without loss of generality" here? Does this mean that other metrics and \gamma values will give same results?

---

> ### Author Response · Authors · 2020-11-14
> **Rebuttal to AnonReviewer4**
>
> **Comment**: How can you justify the proposed approach which measures domain similarity in terms of distances between meta-features (some kind of point estimation), rather than inter-distributional distances? My concern is that the information may be considerably dropped than measuring similarity by inter-distributional distance.
>
> *Rebuttal*: To the best of our knowledge, in all the HPO literature there is no such inter-distributional distance being used. The problem here is especially, that the instances from different datasets will have different predictors, we are not sure that interdistributional distances can be easily applied to such a scenario.
>
> **Comment**: The algorithm in the appendix doesn't have a section on learning meta features, how (and when) are these updated in the algorithm?
>
> *Rebuttal*: We added Algorithm 5 to describe the process of learning meta-features, we also added a new experiment that uses D2V meta-features, a variant of the TST approach, which we call TST-R For our method, we learn the meta-features end-to-end by optimizing Equation 8, page 5. We also present in the Ablation Study, Section 5.5, the effect of pretraining the meta-feature extractor.
>
> **Comment**: Although the range of \hat{s} is defined as {0, 1}, in (8) \hat{s} appears to be a continuous value. Are you using some kind of threshold to make it binary?
>
> *Rebuttal*: \hat{s} is the unnormalized probability for s being 1, thus the correct target space is \R^+_0, not \{0,1\}. We fixed the typo.
>
> **Comment**: The authors argue that "Without loss of generality, we use the Euclidean distance to measure the similarity between the extracted meta features, ... and \gamma = 1". What do you mean by "without loss of generality" here? Does this mean that other metrics and \gamma values will give the same results?
>
> *Rebuttal*: By “without loss of generality” we meant that in principle any distance metric should work, because the model is learning for the chosen distance metric. But we have no empirical results for other distance measures. We rephrased as “For simplicity, we use the Euclidean instance”.

---

> > ### Comment · AnonReviewer4 · 2020-11-18
> > **Answer to rebuttal**
> >
> > The authors have provided answers to my comments, but I have additional questions about the revised parts of the paper.
> >
> > - The results of Experiment Q2 are quite different from those before and after rebuttal. In particular, in the latter half of the trial when the regret is small enough, the performance is no better than the other methods. Why is this?
> >
> > - Although the authors claim to have achieved SOTA in Experiment Q3, the performances of the algorithm (Table 4) are dependent on the hyperparameter K, and failure of tuning can be expected to result in worse performance than the other methods. As R#5 mentioned, I don't think we can say that the proposed method is robust unless  considering the proper way to choose K.
> >
> > - In ablation study, the authors said that the model underfit when all loss is simultaneously used with respect to Optimization Md, is it possible to explain this reason?
> >
> > - In appendix D, the response model parameter of Algorithm 1 and the meta-feature parameter of Algorithm 5 use the same notation, \beta. Are they controlled by the completely same parameters in this paper? In appendix B it says "we incorporate the meta-feature extractor as part of the response model", which confuses me, as if the parameters are not exactly the same.

---

> > > ### Author Response · Authors · 2020-11-25
> > > **Second Rebuttal to AnonReviewer4**
> > >
> > > **Comment**: The results of Experiment Q2 are quite different
> > >
> > > *Rebuttal*: The results of the experiments for Q2 have changed for two reasons. First, we updated all of our experiments based on the new unified architecture to improve the robustness of the model. Second, there was a bug in the GP training code which we resolved in this updated draft. For warm-starting probabilistic approaches such as GP, it is expected that all the models will converge after a number of trials, however we are interested in early gains that are provided by our proposed hyper-parameters compared to other warm-start techniques.
> > >
> > > **Comment**: As R#5 mentioned, I don't think we can say that the proposed method is robust unless considering the proper way to choose K.
> > >
> > > *Rebuttal*:  To address your concerns, we evaluated the performance of GROSI(+X) on 5 different k values on the held-out validation datasets, Appendix E.2, and updated Table 4 with GROSI(+10) since it had on average the best validation performance.  We would like to note that top-k was proposed as a way to introduce uncertainty to our model. Unlike probabilistic surrogates that leverage  uncertainty in the acquisition function, selecting the hyper-parameter from the top-k uniformly at random is a heuristic that guarantees exploration.
> > >
> > > **Comment**: In ablation study, the authors said that the model underfit when all loss is simultaneously used with respect to Optimization Md, is it possible to explain this reason?
> > >
> > > *Rebuttal*: The results have been updated in the ablation study for the pretrained meta-feature extractor due to incorrectly loading the weights. In the results of Table 5, the performance of the pretrained model is poor for 2 out of the 3 datasets. After investigating the training losses, we noticed that when optimizing a pretrained model, the loss converges quickly with approximately half the number of iterations compared to when learning from scratch. This indicates that the algorithm gets stuck at a local optima. For Regularization Md, the slight improvement is considered an artifact of the meta-dataset.
> > >
> > > **Comment**: In appendix D, the response model parameter of Algorithm 1 and the meta-feature parameter of Algorithm 5 use the same notation, \beta. Are they controlled by the completely same parameters in this paper? In appendix B it says "we incorporate the meta-feature extractor as part of the response model", which confuses me, as if the parameters are not exactly the same.
> > >
> > > *Rebuttal*: The parameters defined in each algorithm are self-contained, i.e. \beta reported in Algorithm 5 is different from \beta in Equation 8 and Algorithm 4. Algorithm 5 is a stand-alone meta-feature learning algorithm that is implemented to provide learned meta-features for the zero-shot baseline, NN-D2V and the sequential baseline TST-D2V.

---

### Official Review · AnonReviewer5 · 2020-11-04
**Promising idea, but too many weaknesses in current paper**

**Rating:** 4
**Confidence:** 4

**Review:**


The authors of the paper propose a hyperparameter tuning algorithm that uses a simple non-probabilistic model to predict the performance of any new hyperparameter configuration on a target task. They use transfer learning from related tasks as well as learned meta-features for each task, and then simply pick the hyperparamater configuration predicted to perform best by this NN based model.

The authors provide results on 3 meta-datasets, each a slightly different hyperparamter optimization problem applied to the same 120 UCI datasets, thus providing relataed task for each of these three settings. The algorithm can be used in three modes Q1, as a zero-shot algorithm to immediately select the best performing hyperparameters Q2, as an initialization strategy used in the same way as Q1 but followed by a traditional sequential HPO algorithm, and Q3, as a standalone sequential tuning algorithm, by updating the model with any new evaluations, and then using it to suggest the configurations to evaluate next.


#### Comments / Questions
A crucial detail, that is only cursorly mentioned in the main test, is that the authors tune the architecture of their model to be adapted to the specific meta-dataset used (as described in table 5 in appendix D). I think this should be given more attention, having to tune the surrogate hyperparameters itself seems a fairly major limitation of the method and should be discussed. It is not clear exactly how many different architectures were tested, what the search space was, and how well any specific architecture would generalize to new meta-datasets, or even new datasets on the same meta-dataset.

Without more details on how the tuning of the surrogate architecture exactly happened, it does not seem possible to apply this method to other meta-datasets, making the paper incomplete a non-reproducable.

It is also not clear can the model can avoid getting stuck in local minima? I would be curious why this was not a problem with the Q3 experiments without the exploration usually given by a probabilistic model.

Algorithm 3 in the appendix mentions the use of an acquisition function, but the proposed model does not appear to be probabilistic, so it is not really clear what kind of acquisition function would be used here.

Normalized regret used for all results is never defined.

#### Pros
* Important use case/setting (expeciall Q1 and Q2)
* Simple idea, although not described so straighforwardly, I would avoid all the talk of a grey-box method and focus more on simplicity and the idea of providing a good and simple zero-shot NN model that is able to use meta-features in an effective way
* Meta-datasets useful for research are being published with the paper (or at any rate promised to be published upon acceptance)
* The experimental results look fairly competitive (although with some of the shortcomings and open questions I mention in this review)


#### Cons
* The authors tune their model architecture to be adapted to each specific meta-dataset, but this tuning process is not well described, making the method hard to reproduce.
* The algorithm is not very principled from a probabilistic standpoint when used in setting Q3, the used exploitation and exploration trade off is just an heuristic, and not guaranteed to provide good exploration. I would expect it to plateau at a much worse value than bayesian optimization methods. Honestly I would think the paper would be stronger without this use case described, and more more focus given to the results and the zero-shot case.
* The theory sections in the main text are fairly standard and could be shortened a lot, as well as not really needed to explain the method used, what is more relevant is the actual model used, in terms of architecture and tuning. In the current version the architecture is relegated to the appendix and the tuning absent.
* In general the paper is not really self contained, instead a lot of the important details are contained in the appendix. I would also suggest to be clearer about what the main contribution is.
* Any uncertainty quantification (confidence intervals or error bars) in the results is missing, thus it is difficult to assess how reliable the results are.


I think in the current state the paper is not ready to be published. However the idea is simple and promising. I would suggest the authors improve the description of the experimental results, comparing against some more zero-shot baselines. Also either the tuning process of the model itself should be described in more detail in the main text, or not be required (in case a more general architecture can be provided).



#### Typos/Other minor comments
It would be using always use the same term instead of simetimes probabilistic surrogate and other times response model.

The way l_hat and the selection of the surrogate is chosen by max likelihood is a bit convoluted, but as I said above, I don't think the theory sections in the main text add much insight into the method, just direcly describe the loss.

Section 4:
inputting -> taking as input
page 14 first paragraph:
to optimize outer loop -> to optimize *the* outer loop
Appendix B1:  learningJomaa -> missing space
Appendix Q1: typos in "but the time training required time of underlying model."

---

> ### Author Response · Authors · 2020-11-14
> **Rebuttal to AnonReviewer5 [1/2]**
>
> **Comment**: A crucial detail, that is only cursorly mentioned in the main test, is that the authors tune the architecture of their model to be adapted to the specific meta-dataset ... I think this should be given more attention ...
>
> *Rebuttal*: In our understanding the capacity and architecture of the surrogate models is dependent on the configuration space (hence, meta-dataset dependent), because the complexity of the interaction between the hyper-parameters of a ML method differs from the interaction of the hyper-parameters of other ML methods. In the submission version of the paper, we conducted a search for the best architecture for every meta-dataset based on the best validation score on the respective held-out validation meta-dataset. To be clear we divided each meta-dataset into training, validation and test splits of the meta-tasks and tuned the architecture of the surrogate on the validation split.
>
> However, we agree with your point that for generalizing the surrogate architecture to future meta-datasets it is beneficial for practitioners to have a fixed surrogate that is efficient on all meta-datasets. To show that a single unique surrogate architecture is efficient on all meta-datasets we conducted a new experiment by picking the best architecture based on the average validation meta-dataset score for all our meta-datasets. We uploaded a new version of the paper with the new results using a single fixed architecture for all meta-datasets.
>
> The exact protocol of choosing the architecture is detailed in Section 4, page 5 of the new paper draft, as well as Appendix E.
>
> **Comment**: It is also not clear if the model can avoid getting stuck in local minima?
>
> *Rebuttal*: A probabilistic version of the proposed meta-learned surrogate would be really interesting for a future work. We need to research efficient ways of modeling uncertainty with our surrogate neural networks. At the moment, given the lack of a probabilistic neural network, our exploration is ensured by a heuristic acquisition, which does not select the best configuration, but instead a random one among the top-k best.
>
> To show that our method does not get stuck in a local optima, we extend the HPO experiments for more trials (from previously 50 to 75), see Section 5.4, page 8, as well as Table 4. As you can see, not only does our method avoid getting stuck in local optima, but is also better than probabilistic meta-learned surrogates.
>
> **Comment**: Algorithm 3 in the appendix mentions the use of an acquisition function ...
>
> *Rebuttal*: We introduce Algorithm 4 to formally define GROSI(+X), the sequential optimization variant of our model.
>
> **Comment**: Normalized regret never defined.
>
> *Rebuttal*: The regret represents the distance between an observed response and the optimal response on a response surface. For hyper-parameter optimization, the meta-datasets are provided beforehand, consequently the optimal response is known. The normalized regret represents the regret when the response surface is normalized between (0,1). We provide a definition in the updated draft. The definition can be found in Section 5.1, page 5
>
> **Comment**: Simple idea, although not described so straighforwardly ...
>
> *Rebuttal*: Thanks for the suggestion, we use the extra page to to improve and elaborate the description of our proposed approach.
>
> **Comment**: The algorithm is not very principled from a probabilistic standpoint when used in setting Q3 ...
>
> *Rebuttal*: We added a new experiment with more trials to inspect whether our method plateaus eventually and we do not see the behavior you suggest (see Figure 2). Instead, we observe that our method converges faster to the best configuration than the baselines, which have a probabilistic Bayesian optimization strategy.
>
> **Comment**: The theory sections in the main text are fairly standard ...
>
> *Rebuttal*: You are right, we updated the text to improve readability.
>
> **Comment**: In general the paper is not really self contained, instead a lot of the important details are contained in the appendix. I would also suggest to be clearer about what the main contribution is.
>
> *Rebuttal*: We agree, based on your suggestions, we will use the extra page to make the paper more self-contained.
>
> **Comment**: Any uncertainty quantification (confidence intervals or error bars) in the results is missing ...
>
> *Rebuttal*: We added the uncertainty quantification to the new draft in all the Tables and Figures, as well as the Appendices.

---

> > ### Author Response · Authors · 2020-11-14
> > **Rebuttal to AnonReviewer5 [2/2]**
> >
> > **Comment**: I think in the current state the paper is not ready to be published ...
> >
> > *Rebuttal*:  We addressed all of your concerns (new paper draft and new experiments) and we will address the text editing aspects until the end of the rebuttal (streamline the description of the method and experimental protocol). In essence, we addressed your main points: i.e. i) we showed that a single architecture outperforms the baselines, ii) that our method does not plateau despite not modeling uncertainty directly, thanks to conditioning the surrogate on meta-features and a heuristic greedy exploration, iii) offered uncertainty quantifications of results (mean + variance). We also add two new baselines, the average best hyper-parameter for zero-shot transfer learning, and TST-D2V for sequential optimization, a variant of the TST-R that relies on the learned meta-features to measure similarity across datasets.
> >
> > Please notice that this paper sets the new state-of-the-art in transfer learning for HPO (an important problem for the community) by empirically outperforming all the strong baselines we are aware of, in a fair experimental protocol.

---

> > ### Comment · AnonReviewer5 · 2020-11-16
> > **Answer to rebuttal**
> >
> > I thank the authors for their comprehensive changes and rebuttal. I think the changes improve the paper considerably, but I still have some concerns.
> >
> > The improvement vs baselines are fairly minor, and if we add the fact that the description of the experimental protocol and results isn't as clear and streamlined as it could be (although the latest revision is much better in this regard), and not having the source code available, I am not fully confident that the method can be used as a robust hyperparameter tuning method on new settings. I have seen worse, but usually backed by more interesting theoretical ideas.
> >
> > A stronger baseline that should be added is the simple greedy algorithm used in some of the previous zero-shot HPO work, that starts with the best average configuration, and then adds one configuration at a time, such that for each dataset at least one of the selected algorithms is likely to perform well. This encourages a diverse set of configurations, as opposed to just picking the best average ones, which is likely to select very similar hyperparameters
> >
> > I would suggest using standard deviation (or standard error) instead of variance, as it has the same unit as the mean. In the graph approximate confidence intervals mean +- 1.96 * std dev / sqrt(N) can be used, if I understand your setup correctly in your setup you should have N = 24
> >
> > My concern with the heuristic (picking at random from the top k), is that the idea is not even so well defined. Given a continuous space all top k points would be one infinitesimal distance from each other. This means that it only makes sense (and provide something different from the best predicted configuration) if the space is discretized, but then the question is how does one discretize the space, and how does one chose k. And why would that work robustly? Now to be clear, bayesian methods have their own issues, and often it's better to have good empirical results than theoretical guarantees based on dubious assumptions, but ideally I would need more discussion and stronger results to be convinced that such a heuristic would work robustly.
> >
> > Lastly, I think the exposition is still a little chaotic, for an idea that is fairly simple, and some of the changes seem a little unpolished.
> >
> > Minor typos:
> > 4.1: fixed-datset -> fixed-dataset
> > 5.1: Ther -> The

---

> > > ### Author Response · Authors · 2020-11-17
> > > **Second Rebuttal to AnonReviewer5**
> > >
> > > We thank you again for your quick response and valuable feedback!
> > >
> > > **Comment**: The improvement vs baselines are fairly minor, and if we add the fact that the description of the experimental protocol and results isn't as clear and streamlined as it could be (although the latest revision is much better in this regard), and not having the source code available, I am not fully confident that the method can be used as a robust hyperparameter tuning method on new settings.
> > >
> > > *Rebuttal*: We have uploaded a working copy of our repository as well as the datasets, meta-datasets, meta-features, and splits to the supplementary material, and will publish them upon acceptance of the paper.
> > >
> > > **Comment**: A stronger baseline that should be added is the simple greedy algorithm used in some of the previous zero-shot HPO work, that starts with the best average configuration, and then adds one configuration at a time, such that for each dataset at least one of the selected algorithms is likely to perform well.
> > >
> > > *Rebuttal*: Thank you for your suggestion. We have replaced the average best baselines with the average rank, selecting the hyper-parameters with the highest average ranking score across the meta-train datasets. We also added intermediary zero-shot results to Table 3.
> > >
> > > We would like to kindly point out that one of the additional merits of our proposed algorithm that distinguishes it from the rest of the zero-shot baselines is that it provides a well-initialized surrogate that can adapt to any target dataset.
> > >
> > > **Comment**: This encourages a diverse set of configurations, as opposed to just picking the best average ones, which is likely to select very similar hyperparameters I would suggest using standard deviation (or standard error) instead of variance, as it has the same unit as the mean. In the graph approximate confidence intervals mean +- 1.96 * std dev / sqrt(N) can be used, if I understand your setup correctly in your setup you should have N = 24
> > >
> > > *Rebuttal*: Thank you for pointing that out. You are correct, per split, we evaluate on 24 test datasets. The reported results represent the average across 5-splits (total 120 datasets) of the average of the 24 test datasets per split, and the 1x standard deviation of the average across 5 splits. We updated the draft and pointed that out.
> > >
> > > **Comment**: My concern with the heuristic (picking at random from the top k), is that the idea is not even so well defined. Given a continuous space all top k points would be one infinitesimal distance from each other. This means that it only makes sense (and provide something different from the best predicted configuration) if the space is discretized, but then the question is how does one discretize the space, and how does one chose k. And why would that work robustly? Now to be clear, bayesian methods have their own issues, and often it's better to have good empirical results than theoretical guarantees based on dubious assumptions, but ideally I would need more discussion and stronger results to be convinced that such a heuristic would work robustly.
> > >
> > > *Rebuttal*: We define our problem setting for discrete spaces in consistency with the literature. Zero-shot transfer learning solutions for HPO in continuous spaces is an interesting challenge for future work. We introduce top-k, a heuristic acquisition function, as a variant of our top-1 approach due to the lack of uncertainty of our proposed surrogate model. Similar approaches have been seen in Reinforcement Learning settings with \epsilon-greedy policies that guarantee exploration.
> > >
> > > **Comment**: Lastly, I think the exposition is still a little chaotic, for an idea that is fairly simple, and some of the changes seem a little unpolished.
> > >
> > > *Rebuttal*: We thank you again for pointing out the typos. We hope that the current description of the approach is more self-contained and clear for the readers.
> > >
> > > We would like to note out that even though the idea sounds simplistic, we are the first to propose an optimization approach for hyper-parameter optimization that extends the surrogate definition to the joint dataset/hyper-parameter domain, i.e. relies directly on the datasets, and the first to propose the novel regularization technique that proves its importance in the Ablation Study.
> > >
> > > We welcome any further remarks!

---

> > > > ### Comment · AnonReviewer5 · 2020-11-17
> > > > **Clarification on chosen baseline**
> > > >
> > > > Hi, thanks for you swift answer.
> > > >
> > > > The baseline I meant is what is the following simple greedy  algorithm described for example in some of the work by Wistuba
> > > >
> > > > ```
> > > > zero_shot_configs = ()
> > > > K times:
> > > >    zero_shot_configs.append(theta maximasing loss(results + [theta]))
> > > > ```
> > > > where
> > > > loss(zero-shot-configs) = average performance of using the given zero-shot-configs on the training datasets
> > > >
> > > > As for using a discrete search space, GP based bayesian optimization works best with continuous parameter, and most HPO algorithms have no issues working directly in a continuous space. One would think a very high definition discretization is always better, but actually as I described with a discretization that is very finely grained the algorithm would converge towards the top k selected candidates being always the same. It might be that in practice this is not an issue, as you and the results seem to indicate, but it's not obvious to me that it must be the case, and in my view it detract somewhat from the robustness of the method.

---

> > > > > ### Author Response · Authors · 2020-11-18
> > > > > **Implementation of suggested baseline**
> > > > >
> > > > > Once again, your quick reply and valuable feedback are much appreciated.
> > > > >
> > > > > **Comment**: The baseline I meant is what is the following simple greedy algorithm described for example in some of the work by Wistuba …
> > > > >
> > > > > *Rebuttal*: We believe that you are referring to the following baseline [1]. Based on your recommendation, we have added the results of this method to Table 4 in the updated draft.
> > > > >
> > > > > **Comment**: As for using a discrete search space, GP based bayesian optimization works best with continuous parameters, and most HPO algorithms have no issues working directly in a continuous space. One would think a very high definition discretization is always better, but actually as I described with a discretization that is very finely grained the algorithm would converge towards the top k selected candidates being always the same.
> > > > >
> > > > > *Rebuttal*: You are correct, because for most HPO algorithms, the surrogates are defined solely over the domain of hyper-parameter configurations. However, we extend the domain over the datasets themselves, a particular advantage for HPO.
> > > > > Discretization of continuous spaces has also been investigated in [2]. Unfortunately optimizing general black box functions is outside our problem definition, since finding suitable meta-features for black-box functions is not straightforward and constitutes an interesting line of research for future work.
> > > > >
> > > > >
> > > > > We hope that these findings address your concerns, and we welcome any additional feedback.
> > > > >
> > > > >
> > > > > [1] Wistuba, Martin, Nicolas Schilling, and Lars Schmidt-Thieme. "Sequential model-free hyperparameter tuning." 2015 IEEE international conference on data mining. IEEE, 2015.
> > > > >
> > > > > [2] Volpp, Michael, et al. "Meta-Learning Acquisition Functions for Transfer Learning in Bayesian Optimization." arXiv preprint arXiv:1904.02642 (2019).

---

### Author Response · Authors · 2020-11-15
**Updates for the Rebuttal**

We would like to thank all the reviewers for reading our paper in-depth and providing us with new insights and valuable feedback !

Consequently, we updated our paper to address the major concerns and minor points raised. The updated paper emphasizes the following aspects:


- We agree that the paper is not entirely self-contained, with a lot of relevant information pushed to the Appendix. We used the extra page to elaborate on the methodology in order to improve readability and streamline the description of our method.

- We would like to point out that in the original paper, each proposed architecture generalizes well to unseen datasets within the respective search-space (meta-dataset), as proven by the previously reported results. However, the reviewers show concern that the proposed algorithm does not generalize across meta-datasets, which might limit reproducability. Based on your suggestions, we select one architecture which had the best average validation score across all meta-datasets, Section 4 Page 8 and Appendix E, and update all ours results accordingly.

- We introduce an extensive ablation study in Section 5.5 to analyze the different contributions of the proposed learning objective, Equation 8.

- We add three new baselines, *Average Rank* for zero-shot transfer learning (Q1), *TST-D2V* and *SMFO* for the sequential optimization (Q3).

- We uploaded a working version of our code alongside the datasets, meta-datasets, splits, and meta-features to the supplementary material.

Beside these aforementioned points, we address  the individual concerns in the comments.

---

### Author Response · Authors · 2020-11-18
**On the issue of exploration**

For our proposed approach, we have the benefit of conditioning our surrogate model on the dataset itself, by means of the meta-feature extractor, a set-based function that embeds datasets of all shapes and sizes into a fixed-size space. Therefore our surrogate does not start the exploration from scratch, but it already has a prior-indication of where good local optima reside, based on similar tasks/datasets. That can be clearly seen in Figure 3 at the Appendix F2/Q1.

So exploration in the context of dataset-conditioned surrogates is less critical than if you do HPO on a black-box problem, because we already know potential good local optima based on the zero-shot suggestions.

We believe this is the key why our method outperforms all the prior work in the field. The baselines might have better probabilistic explorations, but they do not start the HPO on good initial local optima, which our method does by means of dataset-conditioned surrogate.

---

### Comment · ~Hadi_Samer_Jomaa1 · 2021-02-15
**General**

We would like to once again thank the reviewers for reading our paper and providing us with
valuable feedback.

We would also like to note this paper has been significantly reworked, and is now available as a preprint on Arxiv under the name [Hyperparameter Optimization with Differentiable Metafeatures](https://arxiv.org/abs/2102.03776)

Please use the following bibtex for any citations !

@article{jomaa2021hyperparameter,
  title={Hyperparameter Optimization with Differentiable Metafeatures},
  author={Jomaa, Hadi S and Schmidt-Thieme, Lars and Grabocka, Josif},
  journal={arXiv preprint arXiv:2102.03776},
  year={2021}
}

---

### Decision · Program_Chairs · 2021-01-07
**Final Decision**

**Decision:**

Reject

**Comment:**

The paper provides a transfer learning approach to HPO. It builds and improves upon existing methods of zero-shot HPO where the high level idea is to use the outcomes of hyper-parameters on an offline collection of datasets in order to speed up HPO on a new dataset. On the plus side, the methods provided seem to be novel, and the results seem to be promising. The main issue is the writing and clarity of the paper, making it hard to be certain of the good qualities of the paper. Aggregating the reviews, the details are too spread out between the appendix and main body, the techniques require more motivation behind them, and important details of the experiment are somewhat vague. The authors provided a modified version which is definitely a step in the right direction, however, it does not seem to be enough. I think this is a solid paper based on a promising idea. However, given the almost unanimous agreement about that crucial gap in clarity even after the modified version was uploaded, I recommend rejecting the paper.